# MODEL PATCHING: CLOSING THE SUBGROUP PERFORMANCE GAP WITH DATA AUGMENTATION

**Karan Goel**\*, **Albert Gu**\*, **Sharon Li, Christopher Ré**
Department of Computer Science, Stanford University
`{kgoel,albertgu,sharonli,chrismre}@cs.stanford.edu`

## ABSTRACT

Classifiers in machine learning are often brittle when deployed. Particularly concerning are models with inconsistent performance on specific *subgroups* of a class, e.g., exhibiting disparities in skin cancer classification in the presence or absence of a spurious bandage. To mitigate these performance differences, we introduce *model patching*, a two-stage framework for improving robustness that encourages the model to be invariant to subgroup differences, and focus on class information shared by subgroups. Model patching first models subgroup features within a class and learns semantic transformations between them, and then trains a classifier with data augmentations that deliberately manipulate subgroup features. We instantiate model patching with CAMEL, which (1) uses a CycleGAN to learn the intra-class, inter-subgroup augmentations, and (2) balances subgroup performance using a theoretically-motivated subgroup consistency regularizer, accompanied by a new robust objective. We demonstrate CAMEL's effectiveness on 3 benchmark datasets, with reductions in robust error of up to 33% relative to the best baseline. Lastly, CAMEL successfully patches a model that fails due to spurious features on a real-world skin cancer dataset.

## 1 INTRODUCTION

Machine learning models typically optimize for average performance, and when deployed, can yield inaccurate predictions on important subgroups of a class. For example, practitioners have noted that on the ISIC skin cancer detection dataset (Codella et al., 2018), classifiers are more accurate on images of benign skin lesions with visible bandages, when compared to benign images where no bandage is present (Bissoto et al., 2019; Rieger et al., 2019).

This subgroup performance gap is an undesirable consequence of a classifier's reliance on subgroup-specific features, e.g. spuriously associating colorful bandages with a benign cancer class (Figure 1). A common strategy to side-step this issue is to use manual data augmentation to erase the differences between subgroups, e.g., using Photoshop (Winkler et al., 2019) or image tools (Rieger et al., 2019) to remove markings on skin cancer data before retraining a classifier. However, hand-crafting these augmentations may be impossible if the subgroup differences are difficult to manually express.

Ideally, we would automatically learn the features differentiating the subgroups of a class, and then encourage a classifier to be invariant to these features when making its prediction. To this end, we introduce *model patching*, a framework that encapsulates this solution in two stages:

- *Learn inter-subgroup transformations.* Isolate features that differentiate subgroups within a class, learning inter-subgroup transformations between them. These transformations change an example's subgroup identity but preserve the class label.
- *Train to patch the model.* Leverage the transformations as controlled data augmentations that manipulate subgroup features, encouraging the classifier to be robust to their variation.

In the first stage of model patching (Section 2.1), we learn, rather than specify, the differences between the subgroups of a class. We assume that these subgroups are known to the user, e.g. this is common when users perform error analysis (Oakden-Rayner et al., 2019). Our key insight here is to learn these differences as inter-subgroup transformations that modify the subgroup membership of examples,

while preserving class membership. Applying these semantic transformations as data augmentations in the second stage allows us to generate "imagined" versions of an example in the other subgroups of its class. This contrasts with conventional data augmentation, where heuristics such as rotations, flips, MixUp or CutOut (DeVries & Taylor, 2017; Zhang et al., 2017) are hand-crafted rather than learned. While these heuristics have been shown to improve robustness (Hendrycks et al., 2019), the invariances they target are not well understood. Even when augmentations are learned (Ratner et al., 2017a), they are used to address data scarcity, rather than manipulate examples to improve robustness in a prescribed way. Model patching is the first framework for data augmentation that directly targets subgroup robustness.

The goal of the second stage (Section 2.2) is to appropriately use the transformations to remove the classifier's dependence on subgroup-specific features. We introduce two algorithmic innovations that target subgroup robustness: (i) a subgroup robust objective and; (ii) a subgroup consistency regularizer. Our subgroup robust objective extends prior work on group robustness (Sagawa et al., 2020) to our subgroup setting, where classes and subgroups form a hierarchy (Figure 2 left). Our new subgroup consistency regularizer constrains the predictions on original and augmented examples to be similar. While recent work on consistency training (Hendrycks et al., 2019; Xie et al., 2019) has been empirically successful in constructing models that are robust to perturbations, our consistency loss carries theoretical guarantees on the model's robustness. We note that our changes are easy to add on top of standard classifier training.

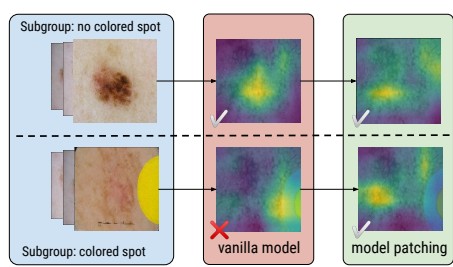

Figure 1: A vanilla model trained on a skin cancer dataset exhibits a subgroup performance gap between images of malignant cancers with and without colored bandages. GradCAM (Selvaraju et al., 2017) illustrates that the vanilla model spuriously associates the colored spot with benign skin lesions. With model patching, the malignancy is predicted correctly for both subgroups.

We contribute a theoretical analysis (Section 3) to motivate our end-to-end framework. Our analysis codifies the distributional assumptions underlying the class-subgroup hierarchy and motivates our new consistency regularizer, which has a simple information theoretic interpretation under this framework. First, we introduce a natural model for the data generating process that decouples an example from its subgroup. Under this model, we prove that our consistency loss in Stage 2, when applied to subgroup augmentations from Stage 1, bounds the mutual information between the classifier output and the subgroup labels. Thus, training with our end-to-end framework forces the classifier to be invariant to subgroup-specific features.

We conduct an extensive empirical study (Section 4) that validates CycleGAN Augmented Model Patching (CAMEL)'s ability to improve subgroup invariance and robustness. We first evaluate CAMEL on a controlled MNIST setup, where it cuts robust error rate to a third of other approaches while learning representations that are far more invariant, as measured by mutual information estimates. On two machine learning benchmarks CelebA and Waterbirds, CAMEL consistently outperforms state-of-the-art approaches that rely on robust optimization, with reductions in subgroup performance gap by up to 10%. Next, we perform ablations on each stage of our framework: (i) replacing the CycleGAN with state-of-the-art heuristic augmentations worsens the subgroup performance gap by 3.35%; (ii) our subgroup consistency regularizer improves robust accuracy by up to 2.5% over prior consistency losses. As an extension, we demonstrate that CAMEL can be used in combination with heuristic augmentations, providing further gains in robust accuracy of 1.5%. Besides CycleGANs, we show that other GAN-based augmentation methods can also be made significantly more robust by combining them with Stage 2 of model patching. Lastly, on the challenging real-world skin cancer dataset ISIC, CAMEL improves robust accuracy by 11.7% compared to a group robustness baseline.

Our results suggest that model patching is a promising direction for improving subgroup robustness in real applications. Code for reproducing our results is available on GitHub.

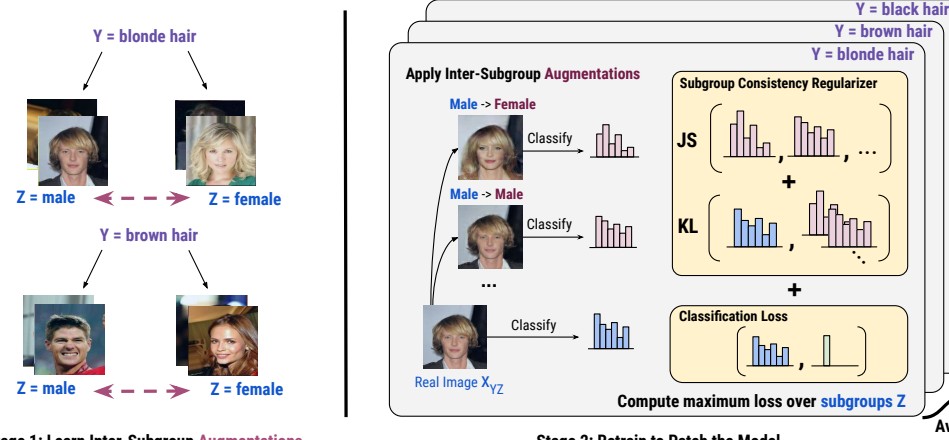

Figure 2: **The model patching framework.** (Left) The class-subgroup hierarchy with each class $Y$ divided into subgroups (e.g. $Y = $ blonde hair into $Z \in \{\text{male}, \text{female}\}$). We learn inter-subgroup augmentations to transform examples between subgroups of a class. (Right) To patch the classifier, we augment examples by changing their subgroup membership and then train with our subgroup consistency loss and robust objective.

## 2 CAMEL: CYCLEGAN AUGMENTED MODEL PATCHING

In this section, we walk through CAMEL's two-stage framework (Figure 2) in detail. In Section 2.1, we introduce Stage 1 of model patching, learning class-conditional transformations between subgroups. In Section 2.2, Stage 2 uses these transformations as black-box augmentations to train a classifier using our new subgroup robust objective (Section 2.2.1) and consistency regularizer (Section 2.2.2). Section 3 outlines our theoretical analysis on the invariance guarantees of our method. A glossary for all notation is included in Appendix A.

**Setup.** We consider a classification problem where $\mathcal{X} \subset \mathbb{R}^n$ is the input space, and $\mathcal{Y} = \{1, 2, \ldots, C\}$ is a set of labels over $C$ classes. Each class $y \in \mathcal{Y}$ may be divided into disjoint subgroups $Z_y \subseteq \mathcal{Z}$[1]. Jointly, there is a distribution $P$ over examples, class labels, and subgroups labels $(X, Y, Z)$. Given a dataset $\{(x_i, y_i, z_i)\}_{i=1}^m$, our goal is to learn a class prediction model $f_\theta : \mathcal{X} \to \Delta^C$ parameterized by $\theta$, where $\Delta^C$ denotes a probability distribution over $\mathcal{Y}$.

### 2.1 STAGE 1: LEARNING INTER-SUBGROUP TRANSFORMATIONS

The goal of the first stage is to learn transformations $F_{z \to z'} : \mathcal{X}_z \to \mathcal{X}_{z'}$ that translate examples in subgroup $z$ to subgroup $z'$, for every pair of subgroups $z, z' \in Z_y$ in the same class $y$.

Recent work has made impressive progress on such cross-domain generative models, where examples from one domain are translated to another, ideally preserving shared semantics while only changing domain-specific features. In this work, we use the popular CycleGAN model (Zhu et al., 2017) to learn mappings between pairs of subgroups, although we show that it is possible to substitute other models (see Section 4.2.4). Given datasets $\{x_z\}_{i=1}^p$, $\{x_{z'}\}_{i=1}^{p'}$ from a pair of subgroups $z, z' \in Z_y$, we train a CycleGAN $F_{z \to z'}$ to transform between them. When classes have more than two subgroups, pairwise models can be trained between subgroups, or multi-domain models such as the StarGAN (Choi et al., 2018) can be used. We include a review of CycleGANs in Appendix C.1.

Given these transformations $\{F_{z \to z'}\}_{z, z' \in Z_y}$, we generate augmented data for every training example $(x, y, z)$ by passing it through all $F_{z \to z'}, z' \in Z_y$. We denote these generated examples $\tilde{x}_{Z_y} := \{\tilde{x}_{z'}\}_{z' \in Z_y}$ where $\tilde{x}_{z'} = F_{z \to z'}(x)$. For convenience, $k$ denotes the number of subgroups $|Z_y|$.

Prior work that uses data augmentation to improve robustness has mostly relied on heuristic augmentations, and focused on robustness to out-of-distribution examples (Hendrycks et al., 2019) with

---

[1]Note that this allows each class to have the same subgroups, or for classes to have overlapping subgroups as special cases. Many of the datasets we consider in our experiments (Section 4) have this property.

Table 1: Comparison of metrics and losses for classifier training. Here $P_z$ and $\hat{P}_z$ are marginal distributions of $(x, y)$ for the subgroup $z$, and $\alpha_\theta(x, y) = \mathbb{I}[(\arg\max f_\theta(x)) = y]$ denotes correct prediction on an example.

|  | Metric of Interest | Loss $\mathcal{L}(\theta)$ |
|---|---|---|
| ERM | $\mathbb{E}_P \alpha_\theta(x, y)$ | $\mathbb{E}_{\hat{P}} \ell(f_\theta(x), y)$ |
| GDRO | $\min_{z \in \mathcal{Z}} \mathbb{E}_{P_z} \alpha_\theta(x, y)$ | $\max_{z \in \mathcal{Z}} \mathbb{E}_{\hat{P}_z} \ell(f_\theta(x), y)$ |
| SGDRO | $\lvert \max_{z \in Z_y} \mathbb{E}_{P_z} \alpha_\theta(x, y) - \min_{z \in Z_y} \mathbb{E}_{P_z} \alpha_\theta(x, y) \rvert$ | $\mathbb{E}_{y \in Y} \{ \max_{z \in Z_y} \mathbb{E}_{\hat{P}_z} \ell(f_\theta(x), y) \}$ |

empirical studies. In contrast, we learn to transform examples rather than specifying augmentations directly, and focus on improving worst-case subgroup robustness. We emphasize that while others have used cross-domain generative models for data augmentation, our novelty lies in targeting invariance to subgroup features using this style of augmentation. Past work has focused on domain adaptation (Huang et al., 2018), few-shot learning (Antoniou et al., 2017), and data scarcity (Bowles et al., 2018; Ratner et al., 2017b), but has not attempted to explicitly control the invariance of the classifier using the learned augmentations. As we describe in our theoretical analysis (Section 3), our use of cross-domain models is a natural consequence of the class-subgroup setting.

## 2.2 STAGE 2: SUBGROUP ROBUSTNESS WITH DATA AUGMENTATION

The goal of the second stage is to learn a classifier $f_\theta$ on both the original and augmented data from Stage 1, using our subgroup robust objective (Section 2.2.1) and consistency regularizer (Section 2.2.2). Our robustness objective targets worst-case subgroup robustness, while our consistency regularizer forces the learned classifier to be invariant to subgroup features. Where relevant, we include discussion here on differences to prior work, with an extended related work in Appendix B.

### 2.2.1 A SUBGROUP ROBUSTNESS OBJECTIVE

We review two established objectives for training classifiers with their associated metrics and loss functions, and introduce our new objective to target subgroup robustness (cf. Table 1).

**Prior work: Empirical Risk Minimization (ERM).** The usual training goal is to maximize the *aggregate accuracy*, optimized using the empirical risk *w.r.t.* a proxy loss function (Table 1, top).

**Prior work: Group Robustness (GDRO).** In our setting, aggregate performance is too coarse a measure of risk, since classes have finer-grained groups of interest. This can be accounted for by optimizing the *worst-case* performance over these groups. Letting $P_z$ denote the conditional distribution of examples associated with subgroup $z \in \mathcal{Z}$, the *robust accuracy* can be quantified by measuring the worst-case performance among all groups. This can be optimized by minimizing the corresponding group robust risk (Table 1, middle right). A stochastic algorithm for this group distributionally robust optimization (GDRO) objective was recently proposed (Sagawa et al., 2020).

**Class-conditional Subgroup Robustness (SGDRO).** The GDRO objective treats group structure as a flat hierarchy. While this approach accounts for worst-case subgroup performance, it loses the class-subgroup hierarchy of our setting. Tailored to this, we create the SGDRO training objective (Table 1, bottom right) to optimize class-conditional worst-case subgroup robustness, aggregated over all classes (Figure 2 right). To measure subgroup robustness, we define the subgroup performance gap (Table 1, bottom left) for a class as the gap between its best and worst performing subgroups.

We note that both GDRO and SGDRO assume knowledge of the subgroups, which is a standard assumption in group robustness (Arjovsky et al., 2019; Ganin et al., 2016; Sagawa et al., 2020).

### 2.2.2 SUBGROUP INVARIANCE USING A CONSISTENCY REGULARIZER

Standard models can learn to rely on spurious subgroup features when making predictions. Subgroup consistency regularization targets this problem by enforcing consistency on subgroup-augmented data, encouraging the classifier to become invariant to subgroup-features.

Recall that Stage 2 connects to Stage 1 by receiving augmented data $\tilde{x}_{Z_y}$, representing "imagined" versions of an example $x$ in all other subgroups $z'$ of its class $y$. We define the *self-consistency* loss $\mathcal{L}_s$ and *translation-consistency* loss $\mathcal{L}_t$ as follows, where $\tilde{m} = \frac{1}{k}\sum_z f_\theta(\tilde{x}_z)$ denotes the average output distribution on the augmented examples.

$$\mathcal{L}_s(x, \tilde{x}_{Z_y}; \theta) = \frac{1}{k}\sum_{z \in Z_y} \mathrm{KL}\left(f_\theta(\tilde{x}_z)\|\tilde{m}\right) \quad (1) \qquad \mathcal{L}_t(x, \tilde{x}_{Z_y}; \theta) = \mathrm{KL}\left(f_\theta(x)\|\tilde{m}\right) \qquad (2)$$

The self-consistency loss is the more important component, encouraging predictions on augmented examples to be consistent with each other. As these augmented examples correspond to one "imagined" example per subgroup, self-consistency controls dependence on subgroup features. Translation consistency additionally forces predictions on the original example to be similar to those of the average CycleGAN-translated examples, ignoring potential artifacts that the CycleGANs generate.

We note that consistency losses have been used before, e.g. UDA (Xie et al., 2019) and Aug-Mix (Hendrycks et al., 2019) use different combinations of KL divergences chosen empirically. Our regularization (1) is tailored to the model patching setting, where it has a theoretical interpretation relating to subgroup invariance (Section 3). We show empirical improvements over these alternate consistency losses in Section 4.2.2.

**Overall Objective.** The total consistency loss averages over all examples,

$$\mathcal{L}_c(\theta) = \frac{1}{2}\mathbb{E}_{(x,y)\sim P}\left[\mathcal{L}_s(x, \tilde{x}_{Z_y}; \theta) + \mathcal{L}_t(x, \tilde{x}_{Z_y}; \theta)\right]. \qquad (3)$$

Combining our SGDRO robust objective and the consistency loss with the consistency strength hyper-parameter $\lambda$ yields the final objective,

$$\mathcal{L}_{\mathrm{CAMEL}}(\theta) = \mathcal{L}_{\mathrm{SGDRO}}(\theta) + \lambda\mathcal{L}_{\mathrm{c}}(\theta). \qquad (4)$$

## 3 AN INFORMATION THEORETIC ANALYSIS OF SUBGROUP INVARIANCE

We introduce a framework to analyze our end-to-end approach, showing that it induces subgroup invariances in the model's features. First, we review a common framework for treating robustness over discrete groups that aims to create *invariances*, or independences between the learned model's features $\phi(X)$ and groups $Z$. We then define a new model for the distributional assumptions underlying the subgroup setting, which allows us to analyze stronger invariance guarantees by minimizing a mutual information (MI) upper bound. Formal definitions and full proofs are deferred to Appendix C.

**Prior work: Class-conditioned Subgroup Invariance.** Prior work (Ganin et al., 2016; Li et al., 2018; Long et al., 2018) uses adversarial training to induce subgroup invariances of the form $(\phi(X) \perp Z) \mid Y$, so that within each class, the model's features $\phi(X)$ appear the same across subgroups $Z$. We call this general approach *class-conditional domain adversarial training* (CDAT). Although these works are motivated by other theoretical properties, we show that they induce the above invariance by minimizing a variational *lower* bound of the corresponding mutual information.

**Lemma 1.** *CDAT minimizes a lower bound on the mutual information $I(\phi(X); Z \mid Y)$.*

Since the model's features matter only insofar as they affect the output, for the rest of this discussion we assume without loss of generality that $\phi(X) = \hat{Y}$ is simply the model's prediction.

**A Natural Distributional Assumption: Subgroup Invariance on Coupled Sets.** Although prior work generally has no requirements on how the data $X$ among the groups $Z$ relate to each other, we note that a common implicit assumption is that there is a "correspondence" between examples among different groups. We codify this distributional assumption explicitly.

Informally, we say that every example $x$ belongs to a *coupled set* $[x]$, containing one example per subgroup in its ($x$'s) class (Figure 3) (Appendix C.3, Definition 1). $[X]$ is the random variable for coupled sets, i.e. it denotes sampling an example $x$ and looking at its coupled set. Intuitively, $x' \in [x]$ represent hidden examples in the world that have identical class features to $x$ and differ only in their subgroup features. These hidden examples may not be present in the train distribution and model patching "hallucinates" them, allowing models to directly learn relevant class features.

This idea of coupled sets underlies both stages of the framework and enables stronger invariance guarantees. Given this notion, all examples $x$ in a coupled set $[x]$ should have identical predictions in order to be robust across subgroups, modeled by the desired invariance $(\hat{Y} \perp Z) \mid [X]$. Instead of Lemma 1, we aim to minimize $I(\hat{Y}; Z \mid [X])$. Note that $I(\hat{Y}; Z \mid [X]) \geq I(\hat{Y}; Z \mid Y)$, which follows from the chain rule for MI (proof in Appendix C), so this is a stronger notion of invariance than CDAT permits. Additionally, the losses from the CycleGAN (Stage 1) and consistency regularizer (Stage 2) combine to form an upper bound on the mutual information rather than a lower bound, so that optimizing our loss is more appropriate.

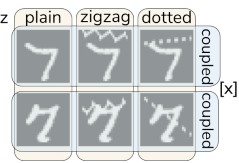

Figure 3: Coupled sets for subgroups of the $Y = 7$ class.

**Theorem 1.** *For a model $f_\theta$ with outputs $\hat{Y}$, the MI $I(\hat{Y}; Z \mid [X])$ is the Jensen-Shannon Divergence (JSD) of predictions on coupled sets $\mathbb{E}_{[x] \sim [X]} JSD \left( f_\theta(x) \right)_{x \in [x]}$. In the case of $k = 2$ subgroups per class, this can be upper bounded by the CycleGAN and consistency losses*

$$\mathbb{E}_{(x,y) \sim (X,Y)} \big( \mathcal{L}_s(x; \tilde{x}_{Z_y}; \theta)^{\frac{1}{2}} + \sum_{z \in Z_y} \mathcal{L}_{CG}^z(x; \theta)^{\frac{1}{2}} \big)^2.$$

*In particular, the global optimum of the trained CAMEL model induces $\hat{Y} \perp Z \mid [X]$.*

The main idea is that the conditional MI $I(\hat{Y}; Z \mid [X])$ can be related to model's predictions on all elements in a coupled set $[x]$ using properties of the JSD. However, since we do not have true coupled sets, the consistency loss (3) only minimizes a proxy for this JSD using the augmentations $\tilde{x}_{Z_y}$. Using standard GAN results, the divergence between the true and augmented distributions can be bounded by the loss of a discriminator, and the result follows from metric properties of the JSD.

Thus, the CycleGAN augmentations (Stage 1) and our consistency regularizer (Stage 2) combine to provide an upper bound on our MI objective, tying together the model patching framework neatly.

## 4 EXPERIMENTS

We demonstrate that CAMEL can take advantage of the learned subgroup augmentations and consistency regularizer to improve robust and aggregate accuracy, while reducing the subgroup performance gap (defined in Table 1). We validate CAMEL against both standard training with no subgroup knowledge (ERM) and other baselines aimed at improving group robustness across 4 datasets. We also conduct extensive ablations to isolate the benefit of the learned inter-subgroup transformations over standard augmentation, and the subgroup consistency regularizer over prior consistency losses.

**Datasets.** We briefly describe the datasets used, with details available in Appendix D.1.

*MNIST-Correlation.* We mix data from MNIST (LeCun et al., 1998) and MNIST-Corrupted (Mu & Gilmer, 2019) to create a controlled setup for analyzing subgroup performance. Digit parity classes $Y \in \{\text{even, odd}\}$ are divided into subgroups $Z \in \{\text{clean, zigzag}\}$ from MNIST and MNIST-Corrupted. $Y$ and $Z$ are highly correlated, so that most even (odd) digits are clean (zigzag).

*CelebA-Undersampled.* Following Sagawa et al. (2020), we classify hair color $Y \in \{\text{non-blonde, blonde}\}$ in the CelebA dataset (Liu et al., 2015). Subgroups are based on gender $Z = \{\text{female, male}\}$. We subsample the set of non-blonde women so that most non-blonde (blonde) examples are men (women).

*Waterbirds.* In this dataset to analyze spurious correlations (Sagawa et al., 2020), birds $Y \in \{\text{landbird, waterbird}\}$ are placed against image backgrounds $Z \in \{\text{land, water}\}$, with waterbirds (landbirds) more commonly appearing against water (land).

*ISIC.* In this skin cancer dataset (Codella et al., 2018), we classify $Y \in \{\text{benign, malignant}\}$ cancers, with bandages $Z$ appearing on $\sim 50\%$ of only benign images.

**Methods.** CAMEL instantiates model patching as described in Section 2. We use the original CycleGAN model with default hyperparameters (Appendix D.2). We compare against ERM and GDRO (Table 1), which respectively minimize the standard risk and robust risk (over all subgroups)

Table 2: A comparison between CAMEL and other methods on 3 benchmark datasets. Evaluation metrics include robust & aggregate accuracy and the subgroup performance gap, calculated on the test set. Results are averaged over 3 trials (one standard deviation indicated in parentheses).

| Dataset | Method | Subgroup Acc. (%) | $Y$ $Z$ | | | Aggregate Acc. (%) | Robust Acc. (%) | Subgroup Gap (%) $Y$ | |
|---------|--------|------|------|------|------|------|------|------|------|
| | | even clean | even zigzag | odd clean | odd zigzag | | | even | odd |
| **MNIST-** | **ERM** | 86.96 | 73.51 | 71.47 | 75.21 | 76.75 (1.60) | 71.47 (1.50) | 13.45 | 3.73 |
| **Correlation** | **IRM** | 94.68 | 69.30 | 81.77 | 93.53 | 84.85 (5.42) | 69.30 (3.29) | 25.38 | 11.76 |
| | **CDAT** | 94.63 | 72.85 | 79.21 | 92.97 | 84.93 (5.84) | 72.85 (3.47) | 21.78 | 13.76 |
| | **GDRO** | 98.10 | 93.31 | 96.82 | 97.15 | 96.35 (0.49) | 93.31 (1.30) | 4.79 | 0.79 |
| | **CAMEL** | 98.85 | 97.89 | 97.98 | 97.87 | **97.55** (0.46) | **97.77** (0.42) | **0.96** | **0.17** |
| | | non-blonde female | non-blonde male | blonde female | blonde male | | | non-blonde | blonde |
| **CelebA-** | **ERM** | 81.09 | 98.08 | 98.13 | 60.04 | 88.26 (1.88) | 62.22 (6.83) | 16.99 | 38.09 |
| **Undersampled** | **GDRO** | 89.26 | 92.24 | 94.08 | 82.20 | 90.91 (0.78) | 82.20 (3.13) | 2.98 | 11.88 |
| | **CAMEL** | 92.15 | 93.73 | 91.13 | 83.53 | **92.90** (0.35) | **83.90** (1.31) | **1.83** | **8.07** |
| | | landbird land | landbird water | waterbird land | waterbird water | | | landbird | waterbird |
| **Waterbirds** | **ERM** | 98.92 | 75.12 | 72.71 | 94.95 | 86.31 (0.39) | 72.71 (2.36) | 23.80 | 22.24 |
| | **GDRO** | 94.46 | 83.81 | 88.19 | 92.36 | 89.39 (0.19) | 83.81 (0.39) | 10.65 | 4.17 |
| | **CAMEL** | 90.84 | 90.40 | 89.69 | 89.58 | **90.89** (0.87) | **89.12** (0.36) | **0.43** | **1.04** |

on the training set. On MNIST-Correlation, we additionally compare against the IRM (Arjovsky et al., 2019) and CDAT (Li et al., 2018) baselines which target invariance assumptions (details in Appendix D.6). All classifiers are fine-tuned using a ResNet-50 architecture, with pretrained ImageNet weights. Detailed information about experimental setups are provided in Appendix D.

**Metrics.** We evaluate on aggregate accuracy, robust accuracy, and subgroup gap for a class $y$, which are the metrics of interest for ERM, GDRO, and our subgroup robustness setting (Table 1).

## 4.1 Subgroup Robustness and Invariance on Benchmark Datasets

We first compare all methods on the benchmark datasets, with results summarized in Table 2.

**CAMEL increases aggregate and robust accuracy while closing the subgroup gap.** On all datasets, CAMEL improves both aggregate and robust accuracy by up to 5.3%, mitigating the tradeoff that other methods experience. CAMEL also balances out the performance of subgroups within each class, e.g., on Waterbirds, reducing this subgroup gap by 10.22% on landbirds compared to GDRO.

**CAMEL learns subgroup-invariant representations.** To measure the invariance of models, we report an estimate of the mutual information defined in Lemma 1, calculated using class-conditional domain prediction heads (Appendix D.5). Table 3 illustrates that CAMEL is the only method that successfully makes the model invariant to subgroups.

Table 3: Estimated MI between predictions and subgroups computed on MNIST-Correlation.

| | ERM | IRM | CDAT | GDRO | CAMEL |
|---|---|---|---|---|---|
| **MI Estimate** | 0.67 | 0.69 | 0.69 | 0.33 | **0.02** |

## 4.2 Model Patching Ablations

We perform ablations on the major components of our framework: (1) substituting learned augmentations with alternatives like heuristic augmentations in Stage 1, and (2) substituting prior consistency losses for our subgroup consistency regularizer in Stage 2.

### 4.2.1 Effect of Learned Augmentations

We investigate the interaction between the type of augmentation used and the strength of consistency regularization, by varying the consistency loss coefficient $\lambda$ on Waterbirds (Table 4). We compare to: (i) *subgroup pairing*, where consistency is directly enforced on subgroup examples from a class without augmentation and (ii) *heuristic augmentations*, where the CycleGAN is substituted with a

Table 4: Ablation analysis (Section 4.2.1) that varies the consistency penalty coefficient $\lambda$. For brevity, we report the maximum subgroup performance gap over all classes.

| Method | Robust Acc. (%) Max Subgroup Gap | | |
|---|---|---|---|
| | $\lambda = 20$ | $\lambda = 50$ | $\lambda = 200$ |
| **Subgroup Pairing** | 74.22 19.53 | 71.88 23.43 | 74.22 23.06 |
| **Heuristic Augmentation** | 87.50 6.95 | 88.54 6.48 | 79.17 37.50 |
| **CAMEL** | 82.03 12.50 | 83.33 10.84 | **89.06** 3.13 |
| **CAMEL + Heuristic** | 89.06 0.21 | **90.62** 1.30 | 53.45 19.39 |

state-of-the-art heuristic augmentation pipeline (Hendrycks et al., 2019) (Appendix D.6) containing rotations, flips, cutout etc. Our goal is to validate our theoretical analysis, which suggests that strong consistency training should help most when used with the coupled examples generated by the CycleGAN. We expect that the ablations should benefit less from consistency training since, (i) subgroup pairing enforces consistency on examples across subgroups that may not lie in the same coupled set; and (ii) heuristic augmentations may not change subgroup membership at all, and may even change class membership.

**Strong consistency regularization enables CAMEL's success.** As $\lambda$ increases from 20 to 200, CAMEL's robust accuracy rises by $7\%$ while the subgroup gap is $9.37\%$ lower. For both ablations, performance deteriorates when $\lambda$ is large. Subgroup pairing is substantially worse ($14.84\%$ lower) since it does not use any augmentation, and as we expected does not benefit from increasing $\lambda$. Heuristic augmentations (e.g. rotations, flips) are not targeted at subgroups and can distort class information (e.g. color shifts in AugMix), and we observe that strongly enforcing consistency ($\lambda = 200$) makes these models much worse. Overall, these results agree with our theoretical analysis.

**CAMEL combines flexibly with other augmentations.** Empirically, we find that using heuristic augmentations in addition to the CycleGAN (method CAMEL + Heuristic) can actually be beneficial, with a robust accuracy of $90.62\%$ and a subgroup gap that is $1.83\%$ lower than using CAMEL alone.

### 4.2.2 ANALYZING THE SUBGROUP CONSISTENCY REGULARIZER

Next, we investigate our choice of consistency regularizer, by substituting it for (i) a triplet Jensen-Shannon loss (Hendrycks et al., 2019) and (ii) a KL-divergence loss (Xie et al., 2019) in CAMEL (Figure 4). Our goal is to show that our theoretically justified regularizer reduces overfitting, and better enforces subgroup invariance.

**Consistency regularization reduces overfitting.** Figure 4 illustrates the train and validation cross-entropy loss curves for CAMEL and GDRO on the small $(\texttt{landbird}, \texttt{water})$ Waterbirds subgroup (184 examples). Consistency regularization shrinks the gap between train and validation losses, strongly reducing overfitting compared to GDRO.

**Alternative consistency losses deteriorate performance.** As expected, substituting the subgroup consistency loss with either the triplet-JS loss or the KL loss in CAMEL reduces robust accuracy significantly ($-2.5\%$ on Waterbirds). Interestingly, our subgroup consistency regularizer improves over prior consistency losses even when used with heuristic augmentations.

Table 5: MNIST-Correlation results when training to predict $Z$ labels and testing on $Y$ labels. Test robust accuracy bolded.

| Method | Subgroup Accuracies | | | |
|---|---|---|---|---|
| **ERM** | 99.18 | 13.18 | **12.24** | 99.46 |
| **GDRO** | 99.02 | **84.00** | 94.20 | 98.78 |

### 4.2.3 TRAINING WITH SUBGROUP INFORMATION

Since subgroup labels are available at train time, a natural baseline to consider is training to predict the finer-grained

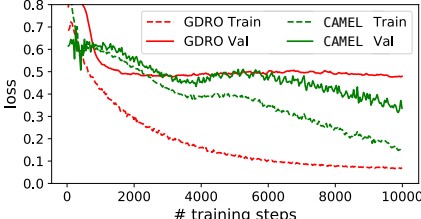

| | Learned Aug. | | Heuristic Aug. | |
|---|---|---|---|---|
| | **Triplet JS** | **KL** | **Triplet JS** | **KL** |
| **Performance Change** (vs. CAMEL Consistency Loss) | -2.50 | -0.83 | -2.08 | -1.04 |

Figure 4: **Consistency loss ablations on Waterbirds**. (Left) loss curves on the $(\text{landbird}, \text{water})$ subgroup. The addition of the CAMEL consistency loss to GDRO reduces overfitting. (Right) Robust accuracy decrease with alternate consistency losses (Triplet JS (Hendrycks et al., 2019) and KL (Xie et al., 2019)) on CAMEL-generated data or heuristic augmentations.

$Z$ label using ERM or GDRO. Evaluation is still performed by comparing against the coarser-grained $Y$ label corresponding to the predicted $Z$ value. We use MNIST-Correlation in the setting of Table 2.

In Table 5, we find that while both ERM and GDRO perform well on the larger subgroups, the performance on minority subgroups drops compared to the simpler setting in Table 2 where they are trained only with $Y$ labels. These results are consistent with the intuition that being asked to predict $Z$ labels forces models to learn information about the spurious attributes instead of the real class, and highlights the importance of erasing subgroup information in order to learn invariant representations.

### 4.2.4 ADDITIONAL GAN ABLATIONS

As the quality of images generated by GANs continues to improve, previous works highlighted in Appendix B have considered them as data augmentation methods, where training an ERM classifier on GAN-augmented images can improve model accuracy. However, in Appendix D.8, we show that this does not improve *robust* accuracy. In contrast, we show that CAMEL can flexibly incorporate three other GAN baselines as alternatives to CycleGAN in Stage 1, whereby combining them with Stage 2 in the model patching pipeline improves robust accuracy by over 20 points.

### 4.3 REAL-WORLD APPLICATION IN SKIN CANCER CLASSIFICATION

We conclude by demonstrating that CAMEL can improve performance substantially on the real-world ISIC (Codella et al., 2018) skin cancer dataset (Table 6). We augment only the benign class, which is split into subgroups due to the presence of a colored bandage (Figure 1) while the malignant class contains no subgroups. We also report AUROC, as is conventional in medical applications.

CAMEL substantially improves robust accuracy by $11.7\%$ and importantly, increases accuracy on the critical malignant cancer class from $65.59\%$ (ERM) and $64.97\%$ (GDRO) to $78.86\%$. While standard ERM models spuriously correlate the presence of the colored bandage with the benign class, CAMEL reduces the model's dependence on spurious features. We verify this by constructing a modified ISIC subgroup (Appendix D.7) for the malignant class that also contains bandages. Figure 1 illustrates using GradCAM (Selvaraju et al., 2017) that CAMEL removes the model's reliance on the spurious bandage feature, shifting attention to the skin lesion instead.

Table 6: Comparison on ISIC. Average of 3 trials (one standard deviation indicated in parentheses).

| Method | Evaluation Metric | |
|---|---|---|
| | **Robust Acc.** | **AUROC** |
| ERM | 65.59 (1.17) | **92.48** (0.80) |
| GDRO | 64.97 (3.15) | 89.50 (2.50) |
| CAMEL | **77.45** (0.35) | 92.47 (0.38) |

## 5 CONCLUSION

Domain experts face a common problem: how can classifiers that exhibit unequal performance on different subgroups of data be fixed? To address this, we introduced model patching, a new framework that improves a classifier's subgroup robustness by encouraging subgroup-feature invariance. Theoretical analysis and empirical validation suggest that model patching can be a useful tool for domain experts in the future.

ACKNOWLEDGMENTS AND DISCLOSURE OF FUNDING

We thank Pang Wei Koh, Shiori Sagawa, Geoff Angus, Jared Dunnmon, and Nimit Sohoni for assistance with baselines and datasets and useful discussions. We thank members of the Hazy Research group including Mayee Chen, Megan Leszczynski, Sarah Hooper, Laurel Orr, and Sen Wu for useful feedback on previous drafts. KG and AG are grateful for Sofi Tukker's assistance throughout this project. We gratefully acknowledge the support of DARPA under Nos. FA86501827865 (SDH) and FA86501827882 (ASED); NIH under No. U54EB020405 (Mobilize), NSF under Nos. CCF1763315 (Beyond Sparsity), CCF1563078 (Volume to Velocity), and 1937301 (RTML); ONR under No. N000141712266 (Unifying Weak Supervision); the Moore Foundation, NXP, Xilinx, LETI-CEA, Intel, IBM, Microsoft, NEC, Toshiba, TSMC, ARM, Hitachi, BASF, Accenture, Ericsson, Qualcomm, Analog Devices, the Okawa Foundation, American Family Insurance, Google Cloud, Swiss Re, the Salesforce Deep Learning Research grant, the HAI-AWS Cloud Credits for Research program, and members of the Stanford DAWN project: Teradata, Facebook, Google, Ant Financial, NEC, VMWare, and Infosys. The U.S. Government is authorized to reproduce and distribute reprints for Governmental purposes notwithstanding any copyright notation thereon. Any opinions, findings, and conclusions or recommendations expressed in this material are those of the authors and do not necessarily reflect the views, policies, or endorsements, either expressed or implied, of DARPA, NIH, ONR, or the U.S. Government.

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

# A    GLOSSARY OF NOTATION

We provide a glossary of notation used throughout the paper.

Table 7: Summary of notation used throughout this work.

| | Notation | Description |
|---|---|---|
| **Preliminaries** | $x, y, z$ | Example, class, subgroup |
| | $X, Y, Z$ | Random variables for examples, classes, and subgroups |
| | $P$ | The joint distribution for $X, Y, Z$ |
| | $P_y, P_z$ | The distribution for $X$ conditioned on class $y$ or subgroup $z$ |
| | $\mathcal{X}, \mathcal{Y}, \mathcal{Z}$ | Domains for $X, Y, Z$ |
| | $Z_y \subset \mathcal{Z}$ | The subgroups belonging to class $y$ |
| | $Y_z \in \mathcal{Z}$ | The class of a subgroup $z$ |
| | $f_\theta : \mathcal{X} \to \Delta^{|\mathcal{Y}|}$ | The parameterized class prediction model, returning a categorical distribution over $\mathcal{Y}$ |
| | $\hat{Y}$ | A random variable with support $\mathcal{Y}$ indicating a random sample from the output of $f_\theta$ |
| **Coupled sets** **and augmentations** | $[x]$ | A coupled set |
| | $[X]$ | Random variable for coupled sets |
| | $[x]_z$ | Example belonging to subgroup $z$ in the coupled set $[x]$ |
| | $x_{Z_y}$ | The coupled set (Definition 1) of examples in $x$'s class $y$. Same as $[x]$. |
| | $[\tilde{x}]$ | An augmented coupled set |
| | $[\tilde{x}]_z, [x]_{\tilde{z}}$ | Example belonging to subgroup $z$ in the augmented coupled set $[\tilde{x}]$ |
| | $\tilde{x}_{Z_y}$ | The augmented coupled set of examples in $\tilde{x}$'s class $y$. Same as $[\tilde{x}]$. |
| | $k$ | Number of subgroups in any (generic) class |
| **Model components** **and losses** | $\mathcal{L}_{CG}$ | Sum of CycleGAN consistency and identity losses |
| | $\mathcal{L}_s$ | Self-consistency loss (Eq 1) |
| | $\mathcal{L}_t$ | Translation-consistency loss (Eq 2) |
| | $\mathcal{L}_c$ | Total consistency loss (Eq 3) |
| | $L : \mathcal{X}^2 \to \mathbb{R}$ | A distance function, used for CycleGAN consistency losses |
| | $\lambda$ | Hyperparameter controlling the strength of the consistency loss |
| | $KL(\cdot)$ | The KL divergence |
| | $JS(\cdot)$ | The Jensen-Shannon divergence (Definition 2) |
| | $I(\cdot)$ | The Mutual Information |

# B    EXTENDED RELATED WORK

We provide a comprehensive overview of related work and highlight connections to our work below.

## B.1    OVERVIEW OF DATA AUGMENTATION

Data augmentation is widely used for improving the aggregate performance of machine learning models in computer vision (Krizhevsky et al., 2012; Szegedy et al., 2014), natural language processing (Kolomiyets et al., 2011; Sennrich et al., 2015; Zhang et al., 2015) and audio (Cui et al., 2015; Ko et al., 2015). The theoretical motivation for data augmentation is largely based on the tangent propagation formalism (Dao et al., 2018; Simard et al., 1991; 1992; 1998) which expresses the desired invariances induced by a data augmentation as tangent constraints on the directional derivatives of the learned model.

Early work considered augmentations as image defects (Baird, 1992) or stroke warping (Yaeger et al., 1996) for character recognition. Since then, augmentation is considered an essential ingredient in computer vision (LeCun et al., 1998; Simard et al., 2003), with commonly used augmentations including random flips, rotations and crops (He et al., 2016; Krizhevsky et al., 2012; Szegedy et al., 2014). Applications of augmentation in computer vision include object detection (Dwibedi et al., 2017; Zoph et al., 2019) and scene understanding (Dvornik et al., 2018)

In natural language processing, common data augmentation techniques include back-translation (Sennrich et al., 2015; Yu et al., 2018), synonym or word substitution (Fadaee et al., 2017; Kobayashi, 2018; Kolomiyets et al., 2011; Wang & Yang, 2015; Zhang et al., 2015), noising (Xie et al., 2017), grammar induction (Jia & Liang, 2016), text editing (Wei & Zou, 2019) and other heuristics (Deschacht & Moens, 2009; Silfverberg et al., 2017). In speech and audio applications, augmentation is also commonly used, through techniques such as vocal tract length warping (Jaitly & Hinton, 2013; Ko et al., 2015) and stochastic feature mapping (Cui et al., 2015; Stylianou et al., 1998).

In this work, we perform an empirical evaluation on image classification tasks although our ideas can be extended to classification of other modalities such as speech and text.

## B.2 Augmentation Primitives and Pipelines

Next, we highlight the particular augmentation primitives that have been used in prior work. Our work is differentiated by the use of learned augmentation primitives using CycleGANs (Zhu et al., 2017), as well as a theoretical justification for this choice.

**Hand-Crafted Augmentation Primitives.** Commonly used primitives are typically heuristic transformations, such as rotations, flips or crops (Krizhevsky et al., 2012; Szegedy et al., 2014). Recent work has hand-crafted more sophisticated primitives, such as Cutout (DeVries & Taylor, 2017), Mixup (Zhang et al., 2017), CutMix (Yun et al., 2019) and MixMatch (Berthelot et al., 2019). While these primitives have culminated in compelling performance gains (Cubuk et al., 2019a;b), they produce unnatural images and distort image semantics.

**Assembling Augmentation Pipelines.** Recent work has explored learning augmentation policies – the right subset of augmentation primitives, and the order in which they should be applied. The learning algorithm used can be reinforcement learning (Cubuk et al., 2019a; Ratner et al., 2017a) or random sampling (Cubuk et al., 2019b). More computationally efficient algorithms for learning augmentation policies have also been proposed (Ho et al., 2019; Lim et al., 2019).

These pipelines are primarily derived from the fixed set of generic image transformations we discussed earlier, and do not directly target specific attributes. By contrast, we consider learning augmentation primitives that target subgroup robustness, and additionally demonstrate in Section 4.2.2 that heuristic augmentations can complement CAMEL to yield additional performance gains.

**Learned Augmentation Primitives.** There is substantial prior work in learning image transformations that produce semantic, rather than superficial changes to an image. A common paradigm is to learn a semantically meaningful data representation, and manipulate embeddings in this representation to produce a desired transformation. Transformations can then be expressed as vector operations over embeddings (Reed et al., 2015; Upchurch et al., 2017) or manifold traversals (Gardner et al., 2015; Reed et al., 2014). Alternative approaches rely on training conditional generative models (Almahairi et al., 2018; Brock et al., 2016; Choi et al., 2018; Isola et al., 2017; Zhu et al., 2017) that learn a mapping between two or more image distributions. Much of this prior work is motivated by the need for sophisticated tools for image editing (Karras et al., 2018; Upchurch et al., 2017) *e.g.* for creative applications of machine learning (Mazzone & Elgammal, 2019).

Closer to our setting is work that explores the use of these transformations for data augmentation. A prominent use case focuses on imbalanced datasets, where learned augmentations are used to generate examples for underrepresented classes or domains. Examples include BaGAN (Mariani et al., 2018), DAGAN (Antoniou et al., 2017), TransferringGAN (Wang et al., 2018) and others (Beery et al., 2019; Hu et al., 2019; Molano et al., 2018; Mounsaveng et al., 2019; Tran et al., 2017; Zhang et al., 2018). Applications to medical data (Pesteie et al., 2019; Sandfort et al., 2019) and person re-identification (chen Sun et al., 2019) have also been explored.

Our model patching framework differs substantially from these papers, since we focus on *robustness*. We discuss this intersection next.

## B.3 Data Augmentation and Model Robustness

Prior work on model robustness has mostly focused on learning models that are robust to bounded $\ell_p$-norm perturbations (Goodfellow et al., 2014b; Moosavi-Dezfooli et al., 2018; Papernot et al., 2015; Szegedy et al., 2013) using ideas such as adversarial training (Madry et al., 2017). A separate line of work considers consistency training (Hendrycks et al., 2019; Kannan et al., 2018; Zheng et al., 2016), where predictions are made invariant to input perturbations, often by minimizing a divergence between the predictions for the original and perturbed examples. Consistency regularization has also been shown to be effective for semi-supervised learning (Xie et al., 2019).

**Consistency training.** We contrast equation (3) with consistency losses from prior work. Unsupervised Data Augmentation (UDA) (Xie et al., 2019) simply controls an asymmetric divergence between the original example and each augmented example individually $\sum_z \text{KL}(f(x)\|f(\tilde{x}_z))$. AugMix (Hendrycks et al., 2019) uses a Jensen-Shannon divergence

$$\frac{1}{k+1}\left[\text{KL}\left(f(x)\|\tilde{m}\right) + \sum_{z \in Z_y} \text{KL}\left(f(\tilde{x}_z)\|\tilde{m}\right)\right]$$

where $\tilde{m} = \frac{1}{k+1}\left[f(x) + \sum_i f(\tilde{x}_i)\right]$. This can be seen as a version of our consistency, but with different weights and a different mean distribution that the KL's are being computed against. Our loss (3) has an important

asymmetry between the original example $x$ and the augmentations $\tilde{x}_i$. One reason to prefer it is simply noting that as the number $k$ of subgroups grows, the AugMix loss tends to the second term, and does not control for the discrepancy between predictions on the original domain $f(x)$ and the augmented ones $f(\tilde{x}_i)$. Our consistency regularization instead allows us to bound a mutual information objective between variables in the joint subgroup distribution, yielding a tractable and interpretable objective (Section 3). In addition, we compare with these consistency losses and provide empirical results in Section 4.2.2.

Robustness to more general augmentations has also been explored (Baluja & Fischer, 2017; Engstrom et al., 2017; Kanbak et al., 2017; Odena et al., 2016; Qiu et al., 2019; Song et al., 2018; Xiao et al., 2018), but there is limited work on making models more robust to semantic data augmentations. The only work we are aware of is AdvMix (Gowal et al., 2019), which combines a disentangled generative model with adversarial training to improve robustness.

Our work contributes to this area by introducing the model patching framework to improve robustness in a targeted fashion. Specifically, under the data-generating model that we introduce, augmentation with a CycleGAN (Zhu et al., 2017) model allows us to learn predictors that are invariant to subgroup identity.

### B.4 Learning Robust Predictors

Recent work (Sagawa et al., 2020) introduced GDRO, a distributionally robust optimization method to improve worst-case accuracy among a set of pre-defined subgroups. However, optimizing the GDRO objective does not necessarily prevent a model from learning subgroup-specific features. Instead, strong modeling assumptions on the learned features may be required, *e.g.* Invariant Risk Minimization (Arjovsky et al., 2019) attempts to learn an invariant predictor through a different regularization term. However, these assumptions are only appropriate for specialized setups where extreme out-of-domain generalization is desired. Unfortunately, these approaches still suffer from standard learning and generalization issues stemming from a small number of examples in the underperforming subgroup(s) – even with perfect subgroup information. Additionally, they necessarily trade off average (aggregate) accuracy against a different robust metric.

### B.5 Relationship to Subgroup Fairness

A common goal in subgroup fairness is to ensure statistical parity of predictions across groups, and a variety of fairness criteria have been proposed (Mehrabi et al., 2019). Others have considered a theoretical setting where the number of subgroups can be large (possibly infinite) (Kearns et al., 2018). Typically, this line of work assumes that groups are common across classes, while in our setting, we consider the possibility of different subgroups in each class. The bulk of this work focuses on prediction problems on small datasets, rather than with high-dimensional image data.

## C Detailed Analysis

We begin with background material on the CycleGAN (Appendix C.1) and the Jensen-Shannon Divergence (Appendix C.2). Appendix C.3 contains a longer discussion of the modeling assumptions in Section 3, fleshing out the distributional assumptions and definition of coupled sets. Appendix C.4 and Appendix C.5 completes the proofs of the results in Section 3.

### C.1 Background: CycleGAN

Given two groups $A$ and $B$, CycleGAN learns mappings $F : B \to A$ and $G : A \to B$ given unpaired samples $a \sim P_A, b \sim P_B$. Along with these generators, it has adversarial discriminators $D_A, D_B$ trained with the standard GAN objective, i.e. $D_A$ distinguishes samples $a \sim P_A$ from generated samples $F(b)$, where $b \sim P_B$. In CAMEL, $A$ and $B$ correspond to data from a pair of subgroups $z, z'$ of a class.

CycleGAN uses a *cycle consistency loss* to ensure that the mappings $F$ and $G$ are nearly inverses of each other, which biases the model toward learning meaningful cross-domain mappings. An additional *identity loss* is sometimes used which also encourages the maps $F, G$ to preserve their original domains *i.e.* $F(a) \approx a$ for $a \sim P_A$. These cycle consistency and identity losses can be modeled by respectively minimizing $\mathcal{L}_{CG}(a, F(G(a)))$ and $\mathcal{L}_{CG}(a, F(a))$ for some function $\mathcal{L}_{CG}$ which measures some notion of distance on $A$ (with analogous losses for $B$). Figure 5 visualizes the CycleGAN model.

**Definition 1.** *The sum of the CycleGAN cycle consistency $\mathcal{L}_{CG}(a, F(G(a))$ and identity $\mathcal{L}_{CG}(a, F(a))$ losses on domain $A$ is denoted $\mathcal{L}_{CG}^A(a; \theta)$ for overall CycleGAN parameters $\theta$, and similarly for domain $B$. In the context of Stage 1 of model patching, let $\mathcal{L}_{CG}^z(x; \theta)$ denote the loss when the domain is one of the subgroups $z$.*

The original CycleGAN uses the $\ell_1$ distance $L(a, \tilde{a}) = \|a - \tilde{a}\|_1$. However, we note that many other functions can be used to enforce similarity. In particular, we point out that a *pair-conditioned discriminator* $\mathcal{D}\{a, \tilde{a}\} \mapsto [0, 1]^2$

can also be used, which accepts a coupled pair of original and translated examples and assigns a probability to each of being the original example. If the guesses for the true and translated examples are $\mathcal{D}_a$ and $\mathcal{D}_{\tilde{a}}$ respectively, then the distance is $L(a, \tilde{a}) = \max_{\mathcal{D}} \log \mathcal{D}_a + \log(1 - \mathcal{D}_{\tilde{a}}) + \log 2$. To sanity check that this has properties of a distance, note that $L$ decreases as $a, \tilde{a}$ are more similar, as the discriminator has trouble telling them apart.

Intuitively, the discriminator loss is a measure of how similar the original and generated distributions are, which will be used in Section C.5 to prove our main result.

## C.2 BACKGROUND: PROPERTIES OF THE JENSEN-SHANNON DIVERGENCE

We define the Jensen-Shannon divergence (JSD) and its properties that will be used in our method and analysis.

**Definition 2.** *The Jensen-Shannon Divergence (JSD) of distributions* $P_1, \ldots, P_k$ *is* $JS(P_1, \ldots, P_k) = \frac{1}{k} \sum_{i=1}^{k} \mathrm{KL}(P_i \| M)$ *where* $M = \frac{1}{k} \sum_{i=1}^{k} P_i$.

*We overload the* $JS(\cdot)$ *function in the following ways. The JSD of random variables* $X_1, \ldots, X_k$ *is the JSD of their laws (distributions).*

*Additionally, we define the JSD of vector-valued inputs if they represent distributions from context. For example, for a model* $f$ *that outputs a vector representing a categorical distribution,* $JS(f_\theta(x_1), \ldots, f_\theta(x_k))$ *is the JSD of those distributions.*

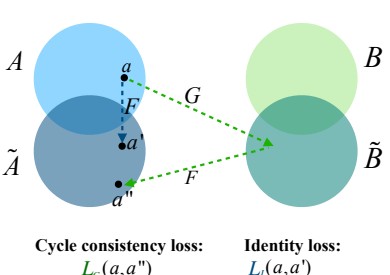

Cycle consistency loss: $L_c(a, a'')$ — Identity loss: $L_i(a, a')$

Figure 5: CycleGAN learns mappings on domains $A \cup B$, where $F$ maps examples to $A$ and $G$ maps to $B$. To model possible distribution shift introduced by the generative model, we denote their images as $\mathrm{Im}(F) = \tilde{A}, \mathrm{Im}(G) = \tilde{B}$ respectively. Semantically consistent mappings are encouraged with the cycle consistency and identity losses, e.g. to ensure that $F(a) = a$ for all $a \in A$.

We briefly review important properties of the JSD. Unlike the KL divergence and other notions of distributional distance, the JSD can be related to a metric.

**Proposition 1.** *The JSD is the square of a metric. In particular, any three distributions* $p, q, r$ *satisfy* $JS(p, q)^{1/2} + JS(q, r)^{1/2} \geq JS(p, r)^{1/2}$.

Finally, the following fact about the JSD relating it to the mutual information of a mixture distribution and its indicator variable will be useful in our analysis.

**Proposition 2.** *Let* $Z$ *be a uniform categorical indicator variable with support* $[k]$ *and* $P_i, i \in [k]$ *be distributions. Let* $X \sim P_z, z \sim Z$ *be the random variable associated with the mixture distribution of the* $P_i$ *controlled by the indicator* $Z$. *Then* $I(X; Z) = JS(P_1, \ldots, P_k)$.

Finally, we review standard results (e.g., from the GAN literature) on the relationship between discriminators and the JS divergence, which relates the loss of an optimal discriminator to the JSD of the two distributions. We include a proof for completeness.

**Proposition 3.** *Consider two domains* $A$ *and* $\tilde{A}$ *(i.e., distributions on a common support* $\mathcal{A}$*), with densities* $p(a), \tilde{p}(a)$ *respectively. Consider a discriminator* $D : \mathcal{A} \to \mathbb{R}$ *optimized to maximize the loss*

$$\mathcal{L}(D) = \frac{1}{2} \mathbb{E}_{a \sim p(a)} \log D(a) + \frac{1}{2} \mathbb{E}_{a \sim \tilde{p}(a)} \log(1 - D(a)).$$

*Then the value of this loss for the optimal discriminator* $D^*$ *is* $JS(A, \tilde{A}) - \log 2$.

*Proof.* Differentiate the loss with respect to the discriminator's output $D(a)$ for any example $a \in \mathcal{A}$, which yields

$$\frac{1}{2} p(a) \frac{1}{D(a)} - \frac{1}{2} \tilde{p}(a) \frac{1}{1 - D(a)}.$$

The loss is maximized at $D^*(a) = \frac{p(a)}{p(a)+\tilde{p}(a)}$. The result follows from plugging this discriminator into the loss and using Definition 2:

$$\begin{aligned}
\mathcal{L}(D^*) &= \frac{1}{2}\mathbb{E}_{a\sim p(a)} \log \frac{p(a)}{p(a)+\tilde{p}(a)} + \frac{1}{2}\mathbb{E}_{a\sim\tilde{p}(a)} \frac{\tilde{p}(a)}{p(a)+\tilde{p}(a)} \\
&= \frac{1}{2}KL\left(A \| \frac{A+\tilde{A}}{2}\right) + \frac{1}{2}KL\left(\tilde{A} \| \frac{A+\tilde{A}}{2}\right) \\
&\quad - \log(2) \\
&= JS(A, \tilde{A}) - \log 2.
\end{aligned}$$

$\square$

### C.3 SUBGROUP INVARIANCE USING COUPLED DISTRIBUTIONS

A common framework for treating robustness over discrete groups aims to create *invariances*, or independencies between the learned model's features and these groups. We review this approach, before defining a new model for the distributional assumptions used in this work. The notion of coupled sets we introduce underlies both stages of the framework and allows for stronger invariance guarantees than previous approaches, which will be analyzed in Appendix C.5.

**Class-conditioned Subgroup Invariance.**  In order for a model to have the same performance over all values of $Z$, intuitively it should learn "$Z$-invariant features", which can be accomplished in a few ways. Invariant Risk Minimization (IRM) (Arjovsky et al., 2019) calls the $Z$ labels *environments* and aims to induce $(Y \mid \phi(X)) \perp Z$, where $\phi(X)$ are the model's features, so that the classifier does not depend on the environment. Another line of work treats $Z$ as *domains* and uses adversarial training to induce invariances of the form $(\phi(X) \perp Z) \mid Y$ (Ganin et al., 2016; Li et al., 2018; Long et al., 2018), so that within each class, the model's features look the same across domains. We call this general approach *class-conditional domain adversarial training* (CDAT), which attaches a domain $Z$ prediction head per class $Y$, and adopts an adversarial minmax objective so that the featurizer $\phi(X)$ erases $Z$ related information and reduces the model's dependence on $Z$.

**Coupling-conditioned Subgroup Invariance.**  Although previous works generally make no assumptions on how the data $X$ among the groups $Z$ relate to each other, we note that a common implicit requirement is that there is a "correspondence" between examples among different groups. We codify this distributional assumption explicitly with a notion of coupling, which allows us to define and analyze stronger invariances.

In particular, we assume that the underlying subgroups are paired or coupled, so that every example can be translated into the other subgroups. Definition 1 formalizes our distributional notion of *coupled sets*.

**Definition 1.** *For a given distribution $P$, a* coupled set *within class $y$ is a set $\{x_z\}_{z\in Z_y}$ consisting of one example from each subgroup of $y$, where each example has the same probability.[2] A* coupling *for a distribution $P$ on $(X, Y, Z)$ is a partition of all examples in $\mathcal{X}$ into coupled sets. For any example $x \in \mathcal{X}$, let $[x]$ denote its coupled set. Let $[x]_1, \ldots, [x]_k$ denote the elements of a coupled set $[x]$ in a class with $k$ subgroups. Let $[X]$ denote the random variable that samples a coupled set; i.e. taking $[x]$ for a random $x$ sampled from any fixed subgroup $z$.*

Additionally, we say that a distribution is *subgroup-coupled* if it satisfies Definition 1, i.e. it has a coupling.

In the context of subgroups of a class $y$, this assumption entails that every example can be factored into its subgroup and coupled set membership. All examples that are members of a particular coupled set can be thought of as sharing a set of common features that signal membership in the class. Separately, examples that are members of a particular subgroup can be thought to share common features that signal subgroup membership. Together, these two pieces of information identify any example from class $c$.

We represent this assumption by letting the (unobserved) random variable $[X]$ represent the "class identity" of an example $X$, which can be thought of as the class features that aren't specific to any subgroup. Thus, the full generating process of the data distribution $(X, Y, Z, [X])$ consists of independently choosing a coupled set $[X]$ and subgroup $Z$ within a class $Y$, which together control the actual example $X$. Note that $[X]$ and $Z$ are both more fine-grained and thus carry more information than $Y$. This process is illustrated in Figure 6a. Figure 6b illustrates this concept for the MNIST-Corrupted dataset (Mu & Gilmer, 2019). Given a digit class such as

---

[2]Note that this will typically not hold for the training distribution, since some subgroups may be underrepresented, making it much less probable that examples from those subgroups are sampled in a coupled set. However, we are concerned with robustness to a test distribution where the subgroups are of equal importance and equally likely.

$Y = 3$, subgroups correspond to corruptions such as zigzags and dotted lines applied to the digits. A coupled set consists of these corruptions applied to a clean digit.

Definition 1 allows us to reason about the following stronger invariances. Given class $y \in \mathcal{Y}$, every example in subgroup $z \in Z_y$ implicitly has corresponding examples in all subgroups $Z_y$ within its class, and the learned features for each of these coupled sets should be identical in order to equalize performance between subgroups. Thus instead of the weaker goal $(\phi(X) \perp Z) \mid Y$, we use the stronger coupling-conditioned invariance $(\phi(X) \perp Z) \mid Y, [X] = (\phi(X) \perp Z) \mid [X]$.

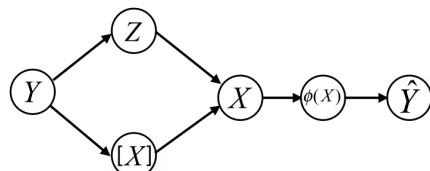

(a) Joint distribution of examples $X$ with their class labels $Y$, subgroup labels $Z$, and coupled sets $[X]$.

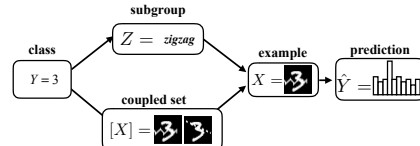

(b) Illustration with the MNIST-Corrupted dataset (Mu & Gilmer, 2019), where subgroups $Z$ are different types of corruptions.

Figure 6: Subgroup-coupled distributions separate the coupled set to which an example belongs (with respect to their class), from its subgroup label.

Note that since features matter insofar as their effect on the final output $\hat{Y}$, it suffices to look at the case $\phi(X) = \hat{Y}$. We first show in Section C.4 that CDAT methods target the invariance $(\hat{Y} \perp Z) \mid Y$ by minimizing a lower bound for the conditional mutual information, $I(\hat{Y}; Z \mid Y)$ (Lemma 1).

In Section C.5, we prove our main result: our combined objective function (4) targets the stronger invariance $(\hat{Y} \perp Z) \mid [X]$ by upper bounding the corresponding MI, which can be interpreted as forcing matching outputs for the examples in every coupled set.

### C.4 MI Bounds for Class-conditioned Invariance

Recall that the high-level goal of CDAT is to induce independencies between subgroup information and the model's feature representation. In order to induce the desired invariance $(\phi(X) \perp Z) \mid Y$ of class features from subgroup identities, a natural approach is to minimize the conditional mutual information $I(\phi(X); Z \mid Y)$, which is minimized at 0 when the invariance is satisfied and grows when $\phi(X)$ and $Z$ are predictive of each other. This mutual information can be estimated using standard techniques.

**Lemma 1.** *CDAT minimizes a lower bound on the mutual information $I(\phi(X); Z \mid Y)$, where $\phi(X)$ is the feature layer where the domain prediction head is attached.*

*Proof.* We have

$$
\begin{aligned}
I(\phi(X); Z \mid Y) &= H(Z \mid Y) - H(Z \mid \phi(X), Y) \\
&= H(Z \mid Y) + \mathbb{E}_{x,y \sim p(x,y)} \mathbb{E}_{z \sim p(z|\phi(x),y)} \left[ \log(p(z|\phi(x),y)) \right] \\
&\geq H(Z \mid Y) + \mathbb{E}_{x,y \sim p(x,y)} \mathbb{E}_{z \sim p(z|\phi(x),y)} \left[ \log(p_\psi(z|\phi(x),y)) \right] \\
&= H(Z \mid Y) + \mathbb{E}_{y,z,\phi(x)} \left[ \log(p_\psi(z|\phi(x),y)) \right],
\end{aligned}
$$

which bounds the MI variationally through a parametrized conditional model $p_\psi$. Up to an additive term $H(Z \mid Y)$ which is a constant of the data distribution, this is simply the cross-entropy loss of a model trained on top of the featurizer $\phi$ to predict $Z$ from $\phi(X)$ and $Y$, which coincides with the domain adversarial training approach. □

By specializing $\phi(X)$ to $\hat{Y}$, we obtain

**Corollary 1.** *If CDAT attaches a domain prediction head to the prediction layer $\hat{Y}$, it optimizes a lower bound on $I(\hat{Y}; Z \mid Y)$.*

Thus, although approaches involving domain adversarial training (Ganin et al., 2016; Li et al., 2018) motivate their approach through alternate concepts such as $\mathcal{H}$-divergences and GAN-based adversarial games, we see that they are implicitly minimizing a simple variational estimate for mutual information.

In Section 4, Table 3's reported estimate of the mutual information uses Corollary 1.

C.5    MI BOUNDS FOR COUPLING-CONDITIONED INVARIANCE

The stronger distributional assumptions of Definition 1 allow us to analyze the invariance $\phi(X) \perp Z \mid [X]$, which can be interpreted as forcing matching features for the data in every coupled set.

**True Coupled Sets.**    Given a subgroup-coupled distribution, access to coupled sets allows analysis of stronger invariance assumptions.

First, we confirm that this is indeed a stronger notion of invariance, that is

$$I(Z; \phi(X) \mid [X]) \geq I(Z; \phi(X) \mid Y). \tag{5}$$

This follows from the chain rule for mutual inequality:

$$\begin{aligned}
I(Z; \phi(X) \mid [X]) &= I(Z; \phi(X) \mid Y, [X]) \\
&= I(Z; [X] \mid Y) + I(Z; \phi(X) \mid Y, [X]) \\
&= I(Z; [X], \phi(X) \mid Y) \\
&= I(Z; \phi(X) \mid Y) + I(Z; [X] \mid Y, \phi(X)).
\end{aligned} \tag{6}$$

Here, the first two equalities follow from Definition 1 (in particular, $[X]$ and $Z$ are more fine-grained than $Y$), and the last two follow from the chain rule for mutual information.

In particular, equation (5) quantifies the intuition that conditioning on an example's coupled set reveals more information then just conditioning on its class. Conversely, minimizing the LHS of (5) necessarily minimizes the objective $I(Z; \phi(X) \mid Y)$ in (Li et al., 2018), and an additional non-negative term $I(Z; [X] \mid \phi(X), Y)$ relating the features and identity of examples.

Moreover, the features $\phi(X)$ are only relevant insofar as their ability to predict the label. Specializing $\phi(X)$, this stronger conditional MI is related to the model's predictions; it is exactly equal to the self-consistency regularizer (1) if the model had access to true coupled sets $[x]$.

Thus, in the case where $\phi(X) = \hat{Y}$ is simply the model's prediction, this MI is simply the Jensen-Shannon divergence of the model's predictions.

**Lemma 2.**

$$I(Z; \hat{Y} \mid [X]) = \mathbb{E}_{[x] \sim [X]} JS\left(f_\theta([x]_1), \ldots, f_\theta([x]_k)\right) \tag{7}$$

*Proof.*    For any features $\phi$, the mutual information can be written

$$\begin{aligned}
I(Z; \phi(X) \mid [X]) &= \mathbb{E}_{[X]} I\left(\mathbb{E}[Z \mid [X]]; \mathbb{E}[\phi(X) \mid [X]]\right) \\
&= \mathbb{E}_{[X]} I\left(Z; \mathbb{E}[\phi(X) \mid [X]]\right)
\end{aligned}$$

where the random variable $\mathbb{E}[\phi(X) \mid [X]]$ denotes the formal conditional expectation. The second equality follows since $(Z \perp [X]) \mid Y$.

Consider specializing this to the case when $\phi(X) = \hat{Y}$, i.e. it represents the random variable where an output class prediction $\hat{Y}$ is sampled from the final class probability predictions $f_\theta(X)$ of the model. Since this is distributed as $P_{\hat{Y} \mid X_z} = f_\theta(X_z)$, we obtain

$$\begin{aligned}
I(Z; \hat{Y} \mid [X]) &= \mathbb{E}_{[x] \sim [X]} \left[ I\left( Z; \frac{1}{k} \sum_{i \in [k]} f_\theta([x]_i) \right) \right] \\
&= \mathbb{E}_{[x] \sim [X]} JS\left(f_\theta([x]_1), \ldots, f_\theta([x]_k)\right)
\end{aligned} \tag{8}$$

where the second equality follows by Proposition 2.    □

**Augmented Coupled Sets.**    In practice, we may not have true coupled sets $[x]$. Instead, we use a generative model such as a CycleGAN as a proxy that provides noisy versions of the coupled set, denoted $[\tilde{x}] = ([\tilde{x}]_1, \ldots, [\tilde{x}]_k)$ where $[\tilde{x}]_i$ are individual augmented examples per subgroup. However, the generative augmentation model may not perfectly model the subgroup distribution; for example, it may introduce artifacts.

We can model this distributional assumption explicitly:

**Definition 3.** *Each subgroup $z$, which has a distribution $P_z$ over $\mathcal{X}$, has a corresponding* augmented subgroup $\tilde{z}$ *with distribution $P_{\tilde{z}}$ representing augmented examples through the generative model(s). In particular, we suppose for any coupled set $[x]$, it has realizations $[x]_z$ in subgroup $z$ and $[\tilde{x}]_z$ in subgroup $\tilde{z}$.*

We also use the notation $[\tilde{x}]$ for a generated coupled set and $[\tilde{x}]_z$ as its realization in subgroup $z$ (a specific augmented example). Note that $[\tilde{x}]$ and the notation $\tilde{x}_{Z_y}$ from Section 2.2 refer to the same thing, the set of augmented examples.

Figure 5 also illustrates the concept of Definition 3: original domains $A, B$ have corresponding domains $\tilde{A}, \tilde{B}$ that are the images of the generators $F, G$.

We can control the difference between augmented and true subgroup distribution in two ways. First, the translation-loss $\mathcal{L}_t$ (2) regularizes the average predictions from the augmentations to match those of the original example, constraining the prediction model to ignore general distribution shifts introduced by the generative models.

Moreover, the discrepancy between the loss we are minimizing via CycleGAN-augmented examples $\mathcal{L}_s = \mathbb{E}_x$ $JS\left(f_\theta([\tilde{x}]_1), \ldots, f_\theta([\tilde{x}]_k)\right)$ (1) and the true objective $JS\left(f_\theta([x]_1), \ldots, f_\theta([x]_k)\right)$ can be bounded by the loss of the pair-conditioned CycleGAN discriminators (Section 2.1), via metric properties of the JSD.

Models such as CycleGAN directly control the deviation of augmentions from the original examples, via the GAN discriminators and consistency losses. The following Lemma says that CycleGAN discriminator loss is the divergence between the original distribution in subgroup $z$, and the generated distribution of subgroup $z$, paralleling standard GAN results (Goodfellow et al., 2014a).

**Lemma 3.** *The optimal discriminator between original subgroup distribution $P_z$ and augmented subgroup $P_{\tilde{z}}$ has loss $\mathcal{L}_{CG}^* = \mathbb{E}_{[x] \sim [X]} JS([x]_z, [\tilde{x}]_z) - \log 2$.*

*Proof of Lemma 3.* By Proposition 3,

$$\mathbb{E}_{[x] \sim [X]} JS([x]_z, [x]_{\tilde{z}}) = \log 2 + \frac{1}{2} \mathbb{E}_{[x] \sim [X]} \log D_{[x]}^z([x]_z) + \frac{1}{2} \mathbb{E}_{[x] \sim [X]} \log(1 - D_{[x]}^z([\tilde{x}]_z))$$

where $D_{[x]}^z$ is a discriminator for this coupled set (within subgroup $z$). Instead of training a separate discriminator per example or coupled set, it is enough to train a single discriminator $D$ conditioned on this specific coupled set $([x]_z, [x]_{\tilde{z}})$. In other words this is a discriminator whose input is both the original example $[x]_z$ and a generated version $[x]_{\tilde{z}}$, and for each input guesses its chance of being a real example. This is exactly the pair-conditioned discriminator described in Section C.1. $\square$

**Proof of Theorem 1.** We finally put the pieces together to prove the main result, restated here for convenience.

**Theorem 1.** *For a model $f_\theta$ with outputs $\hat{Y}$, the MI $I(\hat{Y}; Z \mid [X])$ is the Jensen-Shannon Divergence (JSD) of predictions on coupled sets $\mathbb{E}_{[x] \sim [X]} JSD\left(f_\theta(x)\right)_{x \in [x]}$. In the case of $k = 2$ subgroups per class, this can be upper bounded by the CycleGAN and consistency losses*

$$\mathbb{E}_{(x,y) \sim (X,Y)} \left(\mathcal{L}_s(x; \tilde{x}_{Z_y}; \theta)^{\frac{1}{2}} + \sum_{z \in Z_y} \mathcal{L}_{CG}^z(x; \theta)^{\frac{1}{2}}\right)^2.$$

*In particular, the global optimum of the trained CAMEL model induces $\hat{Y} \perp Z \mid [X]$.*

First, the equivalence of the quantity we care about $I(Z; \hat{Y}; [X])$ and the consistency loss on true coupled sets is given by Lemma 2. It remains to bound $\mathbb{E} JS(f_\theta([x]_1), f_\theta([x]_2))$, which can be bounded by the consistency loss on augmented examples $\mathbb{E} JS(f_\theta([\tilde{x}]_1), f_\theta([\tilde{x}]_2))$ and the optimal CycleGAN losses $\mathbb{E} JS(f_\theta([x]_i), f_\theta([\tilde{x}]_i))$ by metric properties of the JSD.

*Proof of Theorem 1.* Consider any fixed subgroup $z$ and let $\bar{X}_z$ denote the R.V. from the mixture distribution of $P_z$ and $P_{\tilde{z}}$, i.e. either a true example or an augmented example from subgroup $z$. Let $W$ denote the (binary) indicator of this mixture. Then

$$JS(f_\theta([x]_z), f_\theta([\tilde{x}]_z)) = I(W; f_\theta(\bar{X}_z)) \leq I(W; \bar{X}_z) = JS([x]_z, [\tilde{x}]_z), \tag{9}$$

where the equalities are Proposition 2 and the inequality is an application of the *data processing inequality* on the Markov chain $W \to \bar{X}_z \to f_\theta(\bar{X}_z)$.

Combining equation (9) with Lemma 3, applying the definition of $\mathcal{L}_{CG}^z$, and summing over two groups $z = 1, z = 2$ yields

$$JS(f_\theta([x]_1), f_\theta([\tilde{x}]_1))^{\frac{1}{2}} + JS(f_\theta([x]_2), f_\theta([\tilde{x}]_2))^{\frac{1}{2}}$$
$$\leq \mathcal{L}_{CG}^{z_1}(x; \theta)^{\frac{1}{2}} + \mathcal{L}_{CG}^{z_2}(x; \theta)^{\frac{1}{2}} \tag{10}$$

By definition of the self-consistency loss (1) and Definition 2,

$$JS(f_\theta([\tilde{x}]_1), f_\theta([\tilde{x}]_2)) = \mathcal{L}_s(x, [\tilde{x}]; \theta), \tag{11}$$

Table 8: Number of training, validation and test examples in each dataset.

| Dataset | Split | Subgroup Size $(Y, Z)$ | | | |
|---|---|---|---|---|---|
| | | even, clean | even, zigzag | odd, clean | odd, zigzag |
| **MNIST-Correlation** | train | 9900 | 100 | 100 | 9900 |
| | validation | 9900 | 100 | 100 | 9900 |
| | test | 4926 | 4926 | 5074 | 5074 |
| | | landbird, land | landbird, water | waterbird, land | waterbird, water |
| **Waterbirds** | train | 3498 | 184 | 56 | 1057 |
| | validation | 467 | 466 | 133 | 133 |
| | test | 2255 | 2255 | 642 | 642 |
| | | non-blonde, female | non-blonde, male | blonde, female | blonde, male |
| **CelebA-Undersampled** | train | 4054 | 66874 | 22880 | 1387 |
| | validation | 8535 | 8276 | 2874 | 182 |
| | test | 9767 | 7535 | 2480 | 180 |
| | | benign, no bandage | benign, bandage | malignant, no bandage | malignant, bandage |
| **ISIC** | train | 8062 | 7420 | 1843 | 0 |
| | validation | 1034 | 936 | 204 | 0 |
| | test | 1026 | 895 | 239 | 0 |

for any sample $x$ and where $[\tilde{x}]$ denotes the generated coupled set $\{F_1(x), F_2(x)\}$ as usual. Denoting the right hand side $\mathcal{L}_s(x; \theta)$ for shorthand, summing equations (10) and (11), and using the metric property of the JSD (Proposition 1) gives

$$JS(f_\theta([x]_1), f_\theta([x]_2))^{\frac{1}{2}} \leq \mathcal{L}_s(x; \theta)^{\frac{1}{2}} + \mathcal{L}_{CG}^{z_1}(x; \theta)^{\frac{1}{2}} + \mathcal{L}_{CG}^{z_2}(x; \theta)^{\frac{1}{2}}.$$

Finally, squaring and averaging over the dataset and applying Lemma 2 gives the result of Theorem 1:

$$I(\hat{Y}; Z \mid [X]) \leq \mathbb{E}_{x \sim X} \left( \mathcal{L}_s(x; \theta)^{\frac{1}{2}} + \mathcal{L}_{CG}^{z_1}(x; \theta)^{\frac{1}{2}} + \mathcal{L}_{CG}^{z_2}(x; \theta)^{\frac{1}{2}} \right)^2.$$

$\square$

These pieces can be combined to show that the GAN-based modeling of subgroups (Stage 1) and the consistency regularizer (Stage 2) together minimize the desired identity-conditioned mutual information, which completes the proof of Theorem 1.

# D    EXPERIMENTAL DETAILS

We provide detailed information about our experimental protocol and setup for reproducibility, including dataset information in D.1,

## D.1    DATASET INFORMATION

We provide details for preprocessing and preparing all datasets in the paper. Table 8 summarizes the sizes of the subgroups present in each dataset. All datasets will be made available for download.

**MNIST-Correlation.**    We mix data from MNIST (LeCun et al., 1998) and MNIST-Corrupted (Mu & Gilmer, 2019) to create a controlled setup. We classify digit parity $Y \in \{\text{even}, \text{odd}\}$, where each class is divided into subgroups $Z \in \{\text{clean}, \text{zigzag}\}$, drawing digits from MNIST and MNIST-Corrupted (with the zigzag corruption) respectively.

To generate the dataset, we use the following procedure:

- Fix a total dataset size $N$, and a desired correlation $\rho$.
- Sample
    - $\left\lfloor \frac{(\rho+1)N}{4} \right\rfloor$ even digits from MNIST
    - $\frac{N}{2} - \left\lfloor \frac{(\rho+1)N}{4} \right\rfloor$ even digits from MNIST-Corrupted
    - $\frac{N}{2} - \left\lfloor \frac{(\rho+1)N}{4} \right\rfloor$ odd digits from MNIST

   – $\left\lfloor \frac{(\rho+1)N}{4} \right\rfloor$ odd digits from MNIST-Corrupted

This generates a dataset with balanced $Y$ and $Z$ with size $\frac{N}{2}$ each. For our experiments, we use $N = 40000$, $\rho = 0.98$. This makes $Y$ and $Z$ highly correlated, so that most even (odd) digits are clean (zigzag). For validation, we use $50\%$ of the training data.

**CelebA-Undersampled.**  We modify the CelebA dataset (Liu et al., 2015) by undersampling the $(Y = \text{non-blonde}, Z = \text{female})$ subgroup in the training set. The original dataset contains 71629 examples in this training subgroup, and we keep a random subset of 4054 examples. This number is chosen to make the ratio of subgroup sizes equal in both classes $\left(\frac{4054}{66874} \approx \frac{1387}{22880}\right)$. We do not modify the validation or test datasets.

This modification introduces a spurious correlation between hair-color and gender, which makes the dataset more appropriate for our setting. We preprocess images by resizing to $128 \times 128 \times 3$ before use.

**Waterbirds.**  We use the Waterbirds dataset (Sagawa et al., 2020) and resize images to $224 \times 224 \times 3$ before use. Note that this differs from the preprocessing used by (Sagawa et al., 2020), who first resize to $256 \times 256 \times 3$ and then center-crop the image to $224 \times 224 \times 3$. The preprocessing they use makes the task easier, since some part of the (spurious) background is cropped out, while we retain the full image.

**ISIC.**  We use the ISIC dataset (Codella et al., 2018) and resize images to $224 \times 224 \times 3$ before use.

## D.2  CycleGAN Training Details

We use the default hyperparameters suggested by (Zhu et al., 2017) for CycleGAN training, with batchnorm for layer normalization. We use Adam for optimization ($\beta_1 = 0.5$) with a constant learning rate of 0.0002 for both generators and both discriminators.

**MNIST-Correlation.**  Train on 200 images each from both MNIST and MNIST-Corrupted (100 images per class) for 2500 epochs with a batch size of 25, cycle loss coefficient of 10.0 and identity loss coefficient of 1.0. We randomly rotate, pad and crop every image for training. Figure 8 shows some CycleGAN generated examples.

**CelebA-Undersampled.**  Train separate CycleGANs for both classes. Train on 1000 images each from both subgroups within the class for 4000 epochs with a batch size of 16, cycle loss coefficient of 10.0 and identity loss coefficient of 1.0. We flip inputs randomly (with probability 0.5) and randomly crop up to $10\%$ of every image. Due to instability during training, we visually inspected samples generated on the training set at several checkpoints to pick the best model. Figure 9 shows some CycleGAN generated examples.

**Waterbirds.**  Train separate CycleGANs for both classes. Train on 56 and 184 images each from both subgroups for the landbird and waterbird classes respectively. Train for 4000 epochs with a batch size of 4, cycle loss coefficient of 10.0 and identity loss coefficient of 1.0. We flip inputs randomly (with probability 0.5) and randomly crop upto $10\%$ of every image. Figure 10 shows some CycleGAN generated examples.

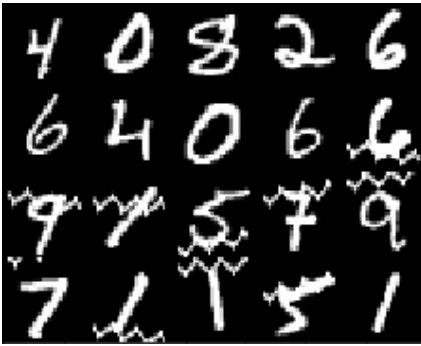

Figure 7: An example of data in MNIST-Correlation. Most even digits are clean while most odd digits contain a zigzag corruption.

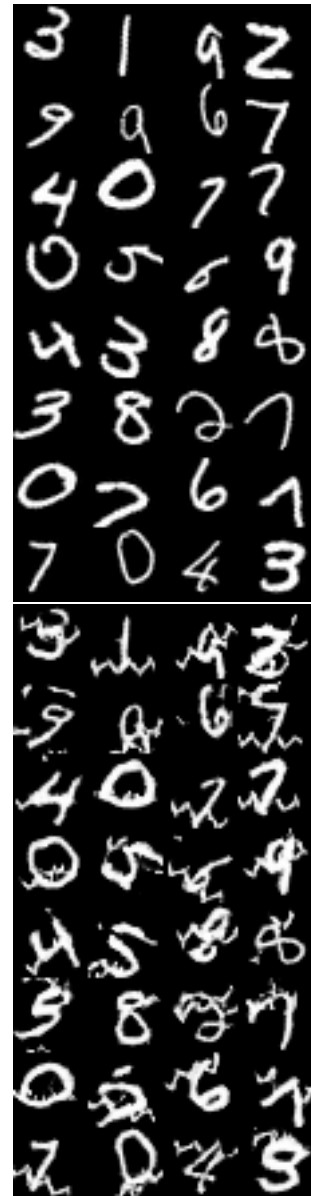

*(top)* real images from MNIST
*(bottom)* CycleGAN generated images

*(top)* real images from MNIST-Correlation
*(bottom)* CycleGAN generated images

Figure 8: Results of inter-subgroup transformations on MNIST-Correlation.

**ISIC.** Train on 100 images each from both benign subgroups (with and without bandaids) for 4000 epochs with a batch size of 4, cycle loss coefficient of 10.0 and identity loss coefficient of 10.0. We flip inputs randomly (with probability 0.5) and randomly crop upto 10% of every image.

### D.3 ARCHITECTURES AND TRAINING INFORMATION

All training code is written in Python with tensorflow-2.0. All models are trained with Stochastic Gradient Descent (SGD), with a momentum of 0.9. In order to isolate the effect of our method, we do not use any data augmentation (such as pad and crop operations or random flips) when training the classifier.

**MNIST-Correlation.** We train a convolutional neural network from scratch, initialized with random weights. The architecture is provided below,

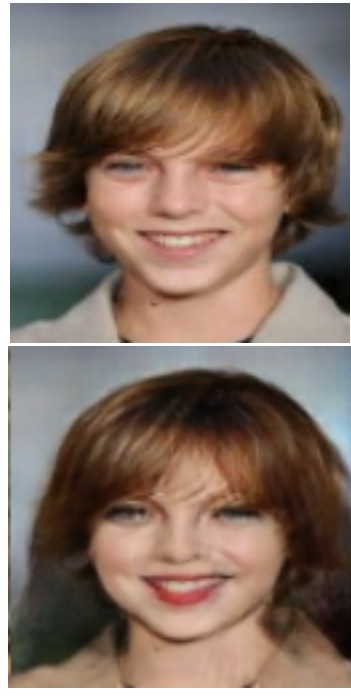
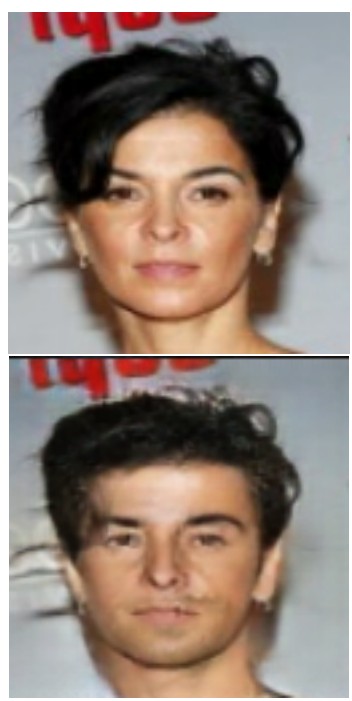

*(top)* male image from CelebA
*(bottom)* CycleGAN generated female image

*(top)* female image from CelebA
*(bottom)* CycleGAN generated male image

Figure 9: Results of inter-subgroup transformations on CelebA-Undersampled. Generation examples use the CycleGAN trained on the non-blonde class.

$\text{Conv2D}(\text{filters}\!=\!32, \text{kernel}\!=\!3) \rightarrow \text{ReLU} \rightarrow \text{Conv2D}(32, 3) \rightarrow \text{ReLU} \rightarrow \text{MaxPooling2D}(\text{pooling}\!=\!2)$
$\rightarrow \quad \text{Dropout}(\text{p}\!=\!0.25) \quad \rightarrow \quad \text{Conv2D}(64, 3) \quad \rightarrow \quad \text{ReLU} \quad \rightarrow \quad \text{Conv2D}(64, 3) \quad \rightarrow \quad \text{ReLU} \quad \rightarrow$
$\text{MaxPooling2D}(2) \rightarrow \text{Dropout}(0.25)$
$\rightarrow \text{Flatten} \rightarrow \text{Dense}(\text{units}\!=\!64) \rightarrow \text{ReLU} \rightarrow \text{Dropout}(0.5) \rightarrow \text{Dense}(10) \rightarrow \text{Softmax}.$

**Other datasets.** All models are fine-tuned using a ResNet-50 architecture, with pretrained ImageNet weights[3]. The only preprocessing common to all methods is standard ImageNet normalization using $\mu = [0.485, 0.456, 0.406], \sigma = [0.229, 0.224, 0.225]$.

### D.4 HYPERPARAMETERS

For model selection, we use robust accuracy on the validation set[4]. The selected model's hyperparameters are then run 3 times, and the results averaged over these trials are reported in Table 2. Below, we provide details of all hyperparameter sweeps, and in Table 12, we include the best hyperparameters found for each method and dataset.

#### D.4.1 CELEBA-UNDERSAMPLED

We run sweeps for all methods over 50 epochs.

**ERM.** Sweep over learning rates $\{0.0001, 0.00005, 0.00002, 0.00001\}$ with weight decay fixed to 0.05.

**GDRO.** Sweep over adjustment coefficients in $\{1.0, 3.0\}$ and learning rates $\{0.0001, 0.00005\}$ with weight decay fixed to 0.05.

---

[3]The particular model used was taken from `https://github.com/qubvel/classification_models`.

[4]For the ISIC dataset, we additionally performed model selection using AUROC, as illustrated in Table 6.

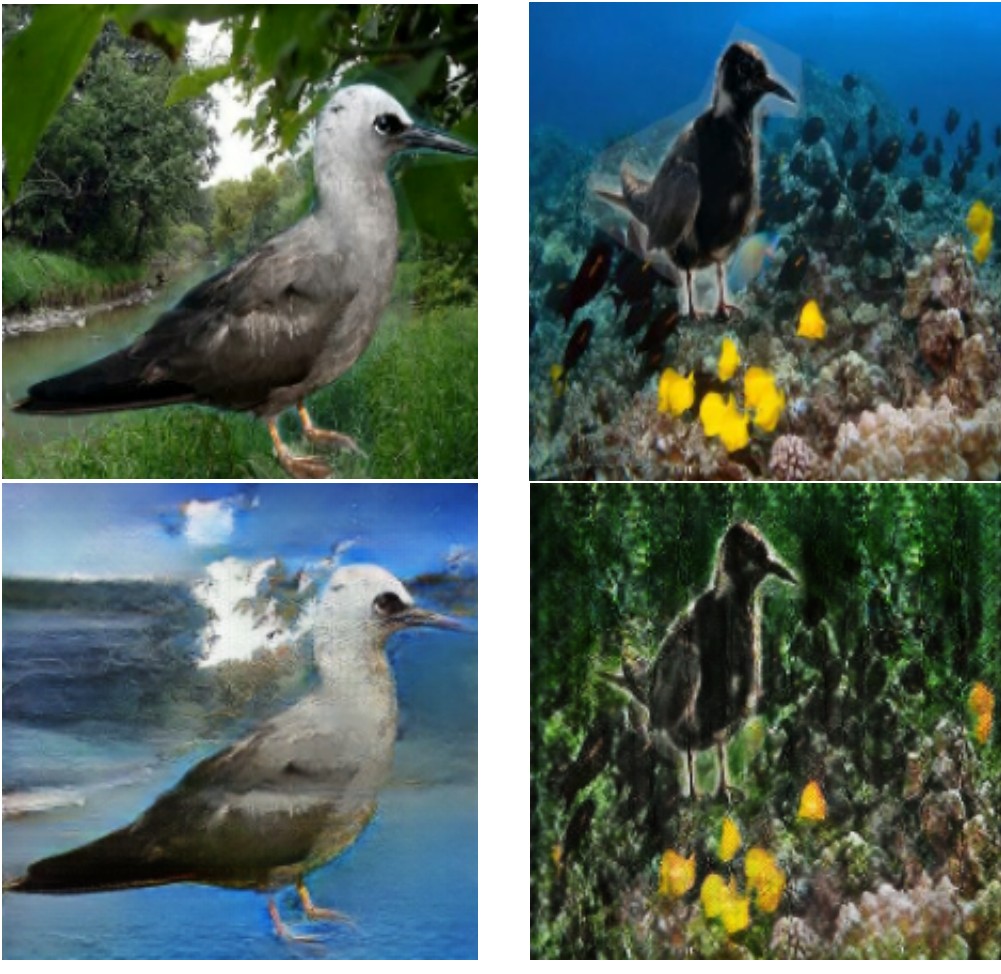

*(top)* land background image from Waterbirds
*(bottom)* CycleGAN generated water background
image

*(top)* water background image from Waterbirds
*(bottom)* CycleGAN generated land background
image

Figure 10: Results of inter-subgroup transformations on Waterbirds. Generation examples use the
CycleGAN trained on the landbirds class.

**CAMEL.** Sweep over consistency penalties in $\{5.0, 10.0, 20.0, 50.0\}$. Learning rate is fixed to $0.00005$, weight decay fixed to $0.05$ and the adjustment coefficient is fixed to $3.0$.

### D.4.2 WATERBIRDS

We run sweeps for all methods over 500 epochs.

**ERM.** Sweep over learning rates $\{0.001, 0.0001, 0.00001\}$ and weight decays $\{0.5, 0.001\}$.

**GDRO.** Sweep over learning rates $\{0.00001, 0.00005\}$ and weight decays $\{0.5, 0.05\}$ with adjustment coefficient fixed to $1.0$ and batch size 24. We also separately swept weight decays $\{1.0, 0.001\}$ and adjustment coefficients over $\{1.0, 2.0\}$.

**CAMEL.** Sweep over consistency penalties in $\{100.0, 200.0\}$ and learning rates $\{0.00005, 0.0001\}$. Weight decay fixed to $0.001$ and adjustment coefficient is fixed to $2.0$. Separately, we sweep over learning rates $\{0.00001, 0.00002, 0.00005, 0.0001\}$, fixing the consistency penalty to $200.0$, weight decay to $0.05$ and adjustment coefficient to $1.0$.

### D.4.3 MNIST-CORRELATION

We run sweeps for all methods over 100 epochs.

**ERM.** Sweep over learning rates $\{0.0001, 0.0002, 0.0005, 0.001\}$ and weight decays $\{0.0005, 0.05\}$.

**GDRO.** Sweep over learning rates $\{0.0001, 0.0002, 0.0005, 0.001\}$ and weight decays $\{0.0005, 0.05\}$. Adjustment coefficient is fixed to $1.0$.

**CDAT.** Sweep over domain loss coefficients $\{-0.1, -0.01, 0.1, 1.0\}$. We fix learning rate to $0.001$ and weight decay to $0.0005$. We run CDAT for $400$ epochs, since it takes much longer to converge.

**IRM.** Sweep over IRM penalty $\{0.01, 0.1, 1.0, 10, 100, 1000, 10000\}$ and learning rates $\{0.0005, 0.001\}$. Weight decay is fixed to $0.0005$.

**CAMEL.** Sweep over consistency penalty weights $\{0.0, 2.0, 5.0, 10.0, 50.0\}$. Learning rate is fixed to $0.001$ and weight decay is fixed to $0.0005$.

### D.4.4 ISIC

We run sweeps for all methods over 75 epochs.

**ERM.** Sweep over weight decays $\{0.5, 0.05, 0.00005\}$. Learning rate is fixed to $0.0001$.

**GDRO.** Sweep over learning rates $\{0.0001, 0.00001\}$ and weight decays $\{0.5, 0.05, 0.00005\}$. Adjustment coefficient is fixed to 0.

**CAMEL.** Sweep over learning rates $\{0.0001, 0.00005\}$, weight decays $\{0.01, 0.05\}$, consistency penalties $\{10.0, 50.0\}$ and annealing rates $\{0.005, 0.002\}$.

### D.5 MUTUAL INFORMATION MEASUREMENT

For the mutual information measurement experiment on MNIST-Correlation in Section 4.1, we additionally attach a domain prediction head to the final feature layer. This domain prediction head is then used to predict the subgroup $z$ of any example $x$. Note that this domain prediction head does not pass back gradients to the main model, it merely observes the learned representation and attempts to improve prediction accuracy of the subgroups using this. Intuitively, this captures how much information about the subgroups is available to be "squeezed-out" by the domain prediction head. This constitutes a use of Lemma 1 to estimate the mutual information, and we report the average cross-entropy loss (added to $\log 2$).

### D.6 BASELINE COMPARISONS

We describe the baselines that we compare to, with implementations for each of these available in our code release.

Table 9: Ablation analysis (Section 4.2.1) that varies the consistency penalty coefficient $\lambda$ on the MNIST-Correlation dataset. For brevity, we report the maximum subgroup performance gap over all classes.

| Method | Robust Acc. (%) Max Subgroup Gap | | | |
| --- | --- | --- | --- | --- |
| | $\lambda = 1$ | $\lambda = 2$ | $\lambda = 5$ | $\lambda = 10$ |
| **Heuristic Augmentation** | 91.90 4.37 | 92.29 3.95 | 92.08 4.94 | 93.22 3.88 |
| **CAMEL** | 97.88 1.16 | 97.02 1.16 | 98.31 1.28 | 98.40 1.21 |

### D.6.1 METHODS

**ERM.** We use standard training with a cross-entropy loss. ERM cannot take advantage of knowledge of the subgroups, so this constitutes a standard baseline that a practitioner might use to solve a task.

**GDRO.** This is our main baseline as described in Section 2, and uses a stochastic optimization method (Sagawa et al., 2020). GDRO uses subgroup information to optimize the worst-case loss over all subgroups. We note that GDRO requires the specification of an adjustment coefficient, and we describe the best found coefficients in Table 12.

**CDAT.** We use a generic domain adversarial training approach using a domain prediction head attached to the last feature layer of the model $\phi(X)$. The domain head predicts the subgroup identity of the given example, and we use gradient reversal in order to erase domain information from the representation $\phi(X)$. We vary the magnitude of the gradient reversal on the domain loss (which we call the domain loss coefficient in Table 12) in order to find the best-performing model.

**IRM.** We implement the IRM penalty (Arjovsky et al., 2019), and treat the subgroups as separate environments across which the model should perform well.

### D.6.2 ABLATIONS

**Subgroup Pairing.** We simply take pairs of examples that lie in different subgroups and enforce consistency on them.

**Heuristic Augmentations.** We build a pipeline inspired by AugMix (Hendrycks et al., 2019) using the following operations: shearing, translation, rotation, flipping, contrast normalization, pixel inversion, histogram equalization, solarization, posterization, contrast adjustment, color enhancement, brightness adjustment, sharpness adjustment, cutout and mixup. We sample between 1 and 3 of these augmentations in a random order and apply them to the image.

We include an additional ablation on the MNIST-Correlation dataset where we vary the consistency penalty coefficient $\lambda$ in Table 9. Compared to heuristic augmentations, CAMEL provides substantial improvements that are stable across different values of $\lambda$.

### D.7 ISIC SPURIOUS CORRELATIONS

For completeness, we include a detailed evaluation for the ISIC dataset in Table 10. Here, we highlight that regardless of what criterion is used for model selection between robust accuracy and AUROC, CAMEL exceeds the performance of the other methods.

For ISIC, we also create an alternate evaluation dataset with artificial images in order to test whether a model spuriously correlates the presence of a bandage with the benign cancer class. To construct this dataset, we use image segmentation to automatically extract images of the bandages from the benign cancer class, and superimpose them on images with malignant cancers. This allows us to generate the artificial subgroup of the malignant cancer class that would contain images with bandages. We use this dataset to highlight how CAMEL improves the model's dependence on this spurious feature in Figure 1.

Table 10: Performance on the ISIC validation set.

| Evaluation Metric | Method | Model Selection Criterion | |
|---|---|---|---|
| | | Robust Acc. | AUROC |
| **Robust Acc.** | ERM | 65.59 (1.17) | 52.93 (10.27) |
| | GDRO | 64.97 (3.15) | 51.23 (1.93) |
| | CAMEL | **77.45** (0.35) | **66.67** (3.03) |
| **AUROC** | ERM | **92.48** (0.80) | **93.38** (0.14) |
| | GDRO | 89.50 (2.50) | 91.83 (0.11) |
| | CAMEL | **92.47** (0.38) | **93.41** (0.52) |

Table 11: Comparisons to GAN Baselines on Waterbirds and CelebA-Undersampled.

| Dataset | GAN Model | Robust/Aggregate Acc. | |
|---|---|---|---|
| | | GAN + ERM | GAN + Model Patching |
| Waterbirds | CycleGAN | 76.88/91.75 | **89.12**/90.89 |
| | Augmented CycleGAN | 63.12/91.08 | **84.87**/86.44 |
| | DAGAN | 73.12/90.28 | — |
| CelebA-Undersampled | StarGAN v2 | 65.91/90.58 | **80.68**/89.33 |

## D.8 ALTERNATIVE GAN AUGMENTATION BASELINES

As noted in Section 2.1, Stage 1 of the model patching pipeline can be integrated with alternative domain translation models. As an additional baseline, we compare to alternative GAN augmentation methods. Typically, these methods are used as a data augmentation method, but not evaluated on robustness.

We consider the Augmented CycleGAN (Almahairi et al., 2018), Data Augmentation GAN (DAGAN) (Antoniou et al., 2017) and StarGAN-v2 (Choi et al., 2020) models, either when used in combination with ERM, or when as a part of the model patching baseline. When used as a part of model patching, we replace the CycleGAN in Stage 1 with the alternative GAN model.

We used released code for Augmented CycleGAN and DAGAN to generate data for the Waterbirds dataset. For StarGANv2, we used pre-trained models for Celeb-A. We note that DAGAN is meant to be a self-contained data augmentation pipeline, so we did not consider it in conjunction with Model Patching.

The results of this comparison is are shown in 11. In particular, these alternate models have poor robust performance when used purely for data augmentation. Their performance improves when integrated in the model patching pipeline.

---

[2]The consistency penalty is increased linearly on every step, from 0 to $\lambda$ with rates 0.002 and 0.005 for $\lambda = 50.0$ and $\lambda = 10.0$ respectively.

Table 12: The values of the best hyperparameters found for each dataset and method.

| Method | Dataset | Hyperparameters | | | | |
|--------|---------|-----------------|---|---|---|---|
| | | **Learning Rate** | **Weight Decay** | **Batch Size** | | |
| **ERM** | **MNIST-Correlation** | 0.0001 | 0.05 | 100 | | |
| | **CelebA-Undersampled** | 0.00005 | 0.05 | 16 | | |
| | **Waterbirds** | 0.001 | 0.001 | 16 | | |
| | **ISIC** | 0.0001 | 0.005 | 24 | | |
| | | 0.0001 | 0.00005 | 24 | | |
| | | **Learning Rate** | **Weight Decay** | **Batch Size** | **GDRO Adjustment** | |
| **GDRO** | **MNIST-Correlation** | 0.0005 | 0.0005 | 100 | 1.0 | |
| | **CelebA-Undersampled** | 0.0001 | 0.05 | 16 | 3.0 | |
| | **Waterbirds** | 0.00001 | 0.05 | 24 | 1.0 | |
| | **ISIC** | 0.0001 | 0.05 | 24 | 0.0 | |
| | | 0.0001 | 0.00005 | 24 | 0.0 | |
| | | **Learning Rate** | **Weight Decay** | **Batch Size** | **GDRO Adjustment** | $\lambda$ |
| **CAMEL** | **MNIST-Correlation** | 0.001 | 0.0005 | 100 | 1.0 | 5.0 |
| | **CelebA-Undersampled** | 0.00005 | 0.05 | 16 | 3.0 | 5.0 |
| | **Waterbirds** | 0.0001 | 0.001 | 16 | 2.0 | 100.0 |
| | **ISIC** | 0.0001 | 0.01 | 24 | 3.0 | 50.0[5] |
| | | 0.0001 | 0.01 | 24 | 3.0 | 10.0[2] |
| | | **Learning Rate** | **Weight Decay** | **Batch Size** | **Domain Loss Coefficient** | |
| **CDAT** | **MNIST-Correlation** | 0.001 | 0.0005 | 100 | -0.10 | |
| | | **Learning Rate** | **Weight Decay** | **Batch Size** | **IRM Anneal Steps** | **IRM Penalty** |
| **IRM** | **MNIST-Correlation** | 0.0005 | 0.0005 | 100 | 2000 | 0.1 |

