# OpenReview forum: "Model Patching: Closing the Subgroup Performance Gap with Data Augmentation"
_ICLR.cc/2021/Conference — ICLR 2021 Poster_

### Official Review · AnonReviewer2 · 2020-10-27

**Rating:** 7
**Confidence:** 3

**Review:**

# Summary

This paper introduces a method (CAMEL) to make CNN models robust to the effect of subgroups in classes. CAMEL uses CycleGAN to transfer the subgroup of each input image in each class and applies consistency regularization among transferred images. This paper additionally introduces a novel objective (SGDRO). CAMEL shows preferable performance on various data, including a real-world dataset of skin cancer classification.

# Strengths

1. The proposed method is straightforward and sound.
1. The authors conducted extensive experiments with various datasets, including a real-world dataset of skin cancer classification. They also performed comprehensive ablation studies to show how and why CAMEL works.
1. CAMEL can be combined with heuristic augmentation (e.g., rotations) to improve robustness.

# Weaknesses

1. The datasets used in the experiments are too simple to show the effectiveness of CAMEL and SGDRO. Specifically, the number of subgroups in each class of all datasets is at most 2. On the other hand, especially SGDRO assumes that subgroups have structure. A more complicated subgroup setting should be considered.
1. When the number of subgroups in each class and the number of classes is limited as the experiments, one can treat $\mathcal{Y}\times\mathcal{Z}$ as target and use ERM. This should be a strong baseline.
1. I concern the scalability of this method. If CycleGANs are used as the paper, $\sum_{y\in\mathcal{Y}}\frac{\|\mathcal{Z}_y\|!}{2}$ CycleGAN models need to be trained on a pair of subgroups beforehand and used during CAMEL training. Even StarGANs are used, $\|\mathcal{Y}\|$ StrarGANs are required.

# Feedbacks

The paper will be improved if generated images of CycleGANs are presented to visually show how CycleGANs change subgroups.

---

> ### Author Response · Authors · 2020-11-21
> **Author Response**
>
> We thank the reviewer for the constructive feedback and discussing important points about the setting and scalability. We particularly appreciate the recommendation of an additional baseline; this was an insightful suggestion that highlights the importance of erasing subgroup information in order to improve invariant representations, and we believe the addition of this ablation strengthens the message of our paper.
>
> We address the reviewer’s comments below:
>
> 1. We believe that two subgroups in a class is sufficient to illustrate the problem of subgroup robustness and demonstrate the potential of the model patching approach. We also note that our choice of datasets directly follows previous work on subgroup robustness [1]. These datasets allow us to validate both stages of model patching on well-established benchmarks. Finally, as noted in our paper, we expect that training a different model such as a StarGAN will seamlessly handle the case with more subgroups at no additional cost.
>
> 2. Reviewer 2 suggests another baseline, which tries to predict Y (class) and Z (subgroup) directly using ERM (note that since our notation defines Z as finer-grained than Y, it is equivalent to predict just Z). Recall that a key hypothesis of model patching is that explicitly modeling Z (subgroup) information makes robustness *more* difficult by reducing invariance to Z in the representation. We therefore expect the suggested baseline to struggle because (i) it must explicitly model Z information, so it cannot learn invariant features (ii) in the data-imbalanced regime, ERM will focus on learning how to predict only the majority subgroups. \
> We confirm this hypothesis with new experiments on the MNIST-Correlation dataset using the same imbalance ratio p=0.99 as in the paper. While ERM achieved >99% test accuracy on the majority subgroups, it actually achieved below average chance on the minority groups, with only 10-20% superclass prediction accuracy. This indicates that it is actually primarily learning how to predict the spurious feature, and not the target class. To sanity check, we ran the Group DRO baseline [1] on the exact same data, which was able to achieve 85-90% test accuracy on the hardest subgroup. \
> Interestingly, this is worse than the 95+% robust accuracy that GDRO achieves in the paper (Table 2) when trained only to predict Y instead of Z. This validates the intuition that explicitly modeling Z (subgroup) information makes robustness more difficult by reducing invariant representations. **We have added a discussion of these new experiments to Section 4.2.3 and Table 5 of our updated draft**.
>
> 3. We agree with the reviewer’s scalability concerns with extremely large |Y|. While addressing high numbers of classes is a difficult area for robustness in general and an active area of research, we note that when the subgroup division is shared among classes, which is a common fairness setting, we could train a *single* GAN to learn subgroup augmentations using all data across classes (optionally, using a class-conditional GAN). In the subgroup robustness setting, the main concern is with respect to sample efficiency instead of computational efficiency, for which our method is very efficient (see responses to Reviewer 1).
>
> 4. We have added examples of generated images for all subgroup augmentations considered in the paper, to visually show how Stage 1 of CAMEL can change subgroups. These can be found in **Figures 8, 9 and 10 of the Appendix in the updated draft**.
>
>
> [1] Sagawa et al. “Distributionally robust neural networks for group shifts:  On the importance of regularization for worst-case generalization.” ICLR 2020.

---

### Official Review · AnonReviewer4 · 2020-10-29
**A model patching based method to design a robust classifier: experiments show reduced subgroup discrepancy on standard datasets**

**Rating:** 7
**Confidence:** 2

**Review:**

Machine learning models are trained to optimize performance on the entire training set, and can often exhibit inaccurate performance on a subgroup. Such inaccuracy often results from the model’s dependence on spurious features. This paper proposes model patching — a two-step method to avoid this problem. The first step learns inter-subgroup transformations where an example from one subgroup is transformed into examples of all the other subgroups within a class. The second stage uses these transformations as controlled data augmentations to learn a classifier that is robust to subgroup-specific variation.

In particular, the paper uses CycleGAN to learn the transformations between pairs of subgroups. The second stage uses the original data and the augmented data from stage 1 and minimizes a subgroup robust objective plus a self-consistency loss. The subgroup robust objective captures the discrepancy between the best and the worst performing subgroup within a class. On the other hand, the self-consistency loss enforces consistency on the augmented data.

Experiments: The authors perform extensive experiments on three benchmark datasets — MNIST, CelebA, and Waterbirds. On all the datasets CAMEL improves both aggregate and robust accuracy by at least 5.3% and also reduces the subgroup gap significantly. Then the authors perform ablations on the two major components of the framework. It seems that learned augmentations perform better than other heuristic augmentations, and substituting the consistency loss with other losses reduces the robust accuracy by 2.5%. Finally, the authors apply the proposed framework on the real-world ISIC skin cancer dataset and found that it improves robust accuracy by 11.7% compared to the other methods.


Strengths:
- Extensive experimentation: I thought that the experiments were sufficient to demonstrate the effectiveness of the proposed approach. In particular, I liked the Model patching ablations experiments, which showed that both the learned augmentations and the subgroup consistency regularizer are important.
- I thought that the use of CycleGAN and SGDRO is well-motivated and appropriate for this setting.

Weaknesses:
- No overlapping subgroups across classes. This paper considers a setting where each class is partitioned into multiple subgroups. I am not sure whether the proposed framework can be easily generalized for the setting when the subgroups are overlapping across multiple classes.



Questions for the authors:
1. What happens if the group information is not known or noisy/incorrect? Can the objective function in subsection 2.2.1 be modified to handle such situations?
2. If I expand the square term given in theorem 1, there will also be a product term in addition to the consistency loss and CycleGAN loss. However, this additional product term is absent from the objective considered by CAMEL. So I did not follow why CAMEL induces the desired conditional independence (\hat{Y} \perp Z | [X]).
3. In table 4, why does CAMEL + Heuristic have very low robust acc (~53%) for \lambda=200? Moreover, for this value of \lambda, the maximum subgroup gap is quite large.
4. Subsection 4.2.2 does not report performance for the datasets — CelebA and MNIST. What is the effect of consistency loss ablations on these datasets?


In summary, this paper proposes a model patching based two-step method to design a classifier that is robust and reduces performance disparity across different subgroups. Experiments on standard datasets clearly show that the proposed method is more effective than some of the existing approaches.

---

> ### Author Response · Authors · 2020-11-21
> **Author Response**
>
> We appreciate the reviewer's detailed review and comments, which raise insightful points about the subgroup setting addressed in this work.
>
> First, we address the comment about overlapping subgroups, which was shared across other reviewers. The setting where each class has the same subgroups or classes have overlapping subgroups is a special case of our method, and our approach of training a class-conditional model to augment between the subgroups applies directly. We note that many of our datasets in fact had this property, for example, both classes of CelebA had subgroups that were male vs. female. We also note that in this special case, a *single* GAN model can be trained in Stage 1 to augment between the common subgroups, regardless of class. **We have added this clarification to Section 2 of our updated draft.**
>
> In response to the four questions listed by the reviewer:
> 1. Misspecification of the finer-grained subgroup labels is a broader challenge for robustness that we do not directly focus on in this work. We point out a few leads: (i) The group DRO algorithm upon which our method is based found that their method is robust to mis-specification of subgroups [1, Appendix B]. (ii) The setting of how to handle subgroup robustness with *no subgroup labels* is an emerging area of robustness, see for example [2] concurrent to this work.
>
> 2. The two terms in Theorem 1 are minimized separately by the two stages of the model patching framework. One term is targeted by the first stage of CycleGAN training, while the other is minimized by the second stage of robust classification. Thus, both stages drive this upper bound down; at the global optimum, if the loss is 0, the overall model then satisfies the desired conditional independence $\hat{Y} \perp Z \mid [X]$.
>
> 3. We attribute the divergence of this particular model (CAMEL+Heuristic, lambda=200) to the fact that heuristic augmentations are not semantically meaningful, so the consistency objective can become unstable with extreme amounts of regularization (high lambda). CAMEL with learned instead of heuristic augmentations did not suffer from this instability, which is evidence for the benefits of our method.
>
> 4. We ran an additional consistency loss ablation for the MNIST-Correlation dataset. We swept lambda={1.0, 2.0, 5.0, 10.0} as reasonable values for this dataset. We found that CAMEL was not sensitive to the consistency penalty and achieved high robustness for all values, with robust accuracy ranging from 97.8%-98.4% and subgroup gap from 1.1-1.3%. On the other hand, the Heuristic augmentation pipeline was substantially less robust, with 91.9%-93.2% robust accuracy and 3.9%-4.9% subgroup gap. \
> One reason why heuristic augmentations are better suited on the Waterbirds dataset is because the spurious features (whether water or land background) are heavily color-based, and heuristic augmentations such as inversion, colorization, solarization, and contrast help remove the dependence on these spurious features. On the other hand, the spurious feature in MNIST-Correlation is the presence of a zigzag, which cannot be removed by heuristic augmentations, but is able to be learned by the CycleGAN in Stage 1 of CAMEL. **This additional ablation can be found in Appendix D.6.2 and Table 9 of our updated draft.**
>
>
> [1] Sagawa et al. “Distributionally robust neural networks for group shifts:  On the importance of regularization for worst-case generalization.” ICLR 2020. \
> [2] Sohoni et al. “No Subclass Left Behind: Fine-Grained Robustness in Coarse-Grained Classification Problems.” NeurIPS 2020.

---

### Official Review · AnonReviewer1 · 2020-10-29
**Intuitive and compelling solution to an interesting problem.**

**Rating:** 7
**Confidence:** 3

**Review:**

The paper focuses on data augmentation in cases when the classifier performance is worse on specific parts of the data. The problem is also closely related to that of the spurious correlation problem, the setting where the classifier might pick up on "random" patterns in the data to make its decision. The solution proposed by the paper is quite intuitive: first, given the subgroups in a class, a CycleGAN model is used to learn different versions of the same training example, each corresponding to a different subgroup. Once the augmented versions of the examples are available, the the classifier is then trained with additional penalty terms, ensuring that the predictions are consistent across different versions of the same training example. Empirical results show that the proposed method does reasonably well as compared to the competitors.

While the reviewer is not an expert in the area, the contribution of the paper indeed sounds appealing. The proposed method is quite intuitive and simple to implement; and the empirical results are quite encouraging. On top of that, the proposed method is quite general (does not seem to be limited to just images) and can be applied to a variety of domains. One drawback of course is the need for manual identification of the subgroups, but given that state-of-the-art methods also need manual annotations (at least to my knowledge), that is probably fine. Finally, the paper is quite well written and easy to follow.

A few comments and suggestion:

- The current assumption is that the subgroups are specific to a single class. However, that may not always be the case in the real world. Consider for instance the problem of fair classification where subgroups (e.g., socially salient groups) might span multiple classes. Does the proposed method extend to such cases? From the first sight, that does not seem to be the case.

- It is not clear what is meant by the statement "handcrafting these augmentations may be impossible if the subgroup differences are difficult to express". However, if the differences are difficult to express, wouldn't it also mean that separating these subgroups is difficult in the first place? In such cases, even the proposed method would also have a problem.

- One potential problem with the proposed approach is its application in domains with small training datasets. It looks like training the CycleGAN would require a relatively large amount of data. How does the proposed approach expected to perform in such cases?

- In Eqs. 2 and 3, the KL divergence is computed between the predicted distributions (presumably, softmax output). Given that DNNs tend to be quite badly calibrated (https://arxiv.org/pdf/1706.04599.pdf), is it worth computing the full KL divergence? Shouldn't minimizing the difference between the argmax class be sufficient?

---

> ### Author Response · Authors · 2020-11-21
> **Author Response**
>
> We are encouraged that the reviewer found our method simple, intuitive, and general. We thank the reviewer for their helpful comments and suggestions, which we address below:
>
> - The setting where each class has the same subgroups or classes have overlapping subgroups is a special case of our method, and our approach of training a class-conditional model to augment between the subgroups applies directly. We note that many of our datasets in fact had this property, for example, both classes of CelebA had subgroups that were male vs. female. We also note that in this special case, a *single* GAN model can be trained in Stage 1 to augment between the common subgroups, regardless of class. **We have added this clarification to Section 2 of our updated draft.**
>
> - The statement "handcrafting these augmentations may be impossible if the subgroup differences are difficult to express" means that it can be difficult to manually define a function to augment the subgroups (e.g., change male to female). However, in this scenario, it is still easy to label the subgroups (i.e., labeling images as male or female), and then an appropriate augmentation can be learned instead of hand-crafted, which is the motivation of our approach. **We have improved the wording of this phrase in Section 1 of our draft.**
>
> - Our approach actually does not require much data. The first stage (learning the augmentations) does not require any additional data compared to the second stage (learning a classifier). For example, our most data-limited dataset **Waterbirds used less than 200 examples per subgroup for each CycleGAN model (see Appendix D.2 Waterbirds)**.
>
> - We found that enforcing the prediction distributions of the original and augmented examples to be close -- in other words, consistency -- is better than directly minimizing the loss to the target output. Intuitively, forcing the model to match the predictions of original and augmented examples (regardless of calibration) more directly enforces the model to ignore subgroup differences, encouraging invariance.

---

### Official Review · AnonReviewer3 · 2020-11-02
**Interesting, well-written paper in an important topic**

**Rating:** 8
**Confidence:** 4

**Review:**

The authors consider the problem  of differential  performance across subgroups commonly present in classifiers, and propose to mitigate it. They augment subgroup data for each class using CycleGAN, and balance the performance across subgroups in each class using a consistency regularizer and a robust objective that minimizes the difference between the minimum and maximum subgroup performance.  This approach is demonstrated in several datasets including the ISIC skin cancer data.

The paper is written well, easy to follow, and I was able to understand and appreciate the contributions quickly. The appendices are also very  thorough and the code is organized well. There is sufficient detail to help readers reproduce the results.

Comments:
1. The authors can specify very early on that the subgroups are pre-specified by the user, and not automatically discovered.
2. Relations to the  area of group fairness may be helpful. The goal there is also  to ensure parity of predictions among groups, and various methods are used. The groups though are assumed to be common across classes. It may be possible to extend this approach to that area, and the idea of data augmentation may be very appealing there. A small discussion on this can be quite informative to the community. There is also the notion of  subgroup fairness discussed here which considers exponentially or infinitely many subgroups
http://proceedings.mlr.press/v80/kearns18a.html
3. Have the authors considered how their methods can be modified when the subgroups are common  across classes (this points directly to the group fairness comment  before)?
4. Just to clarify, the robustness metric used in the experiments is the same as the metric of interest for GDRO in Table 1, correct? And is the gap same as the metric of interest for SGDRO? Does the performance for groups in each class vary between robust accuracy, and robust accuracy + gap? How are the metrics reported in the experiments consolidated across classes? It may be helpful to define the experimental metrics explicitly.
5. Saying that your minimization of I(\hat{Y}; X | [X]) is parallel to Lemma 1 is confusing, since you minimize the upper bound (in Thm. 1) and Lemma 1 minimizes the lower bound. Agree that your minimization is stronger.
6. Have you tried  any preliminary experiments with greater than 2 subgroups per class? Just curious what it takes to use something like StarGAN like you mention.
7. What are the risks of letting the users specify the subgroups? In some problems it may be hard to diagnose which subgroups are meaningful. Is minimizing the worst case performance over many possible automatically discovered subgroups an interesting future  direction in your opinion?  What does it take to do that?
8. How well  will  this method work when there is considerable imbalance in the subgroups considered?
9. In table 5,  what do the quantities in the parenthesis mean?

Typo:
Sec. D.4.2 (GDRO): weighte ->  weight

---

> ### Author Response · Authors · 2020-11-21
> **Author Response**
>
> We are glad the reviewer found the paper well-written and interesting and appreciate their detailed comments and suggestions, which have helped us improve our draft.
>
> We address the comments raised by the reviewer below:
>
> 1. We have **added text on page 1 that clarifies that subgroups are known.**
>
> 2. We appreciate the reviewer’s pointers to the literature. **We have added a small discussion of the group fairness setting in the related work (see Appendix B.5).**
>
> 3. The setting where each class has the same subgroups or classes have overlapping subgroups is a special case of our method, and our approach of training a class-conditional model to augment between the subgroups applies directly. We note that many of our datasets in fact had this property, for example, both classes of CelebA had subgroups that were male vs. female. We also note that in this special case, a *single* GAN model can be trained in Stage 1 to augment between the common subgroups, regardless of class. **We have added this clarification to Section 2 of our updated draft.**
>
> 4. The metrics “Aggregate Accuracy”, “Robust Accuracy”, and “Subgroup Gap” correspond to the defined metrics for ERM, GDRO, and SGDRO respectively in Table 1. The “Max subgroup gap” (Table 4) is the worst-case subgroup gap across all classes. In the revised draft, **we added a “Metrics” paragraph in the beginning of Section 4 to explain our metrics.**
>
> 5. Thanks for pointing this out; **we have improved the wording around Theorem 1.**
> 6. Our datasets were chosen following previous work on group robustness [1], so we did not have experiments with more subgroups per class. However, in Appendix D.8, we showed that a StarGAN can be used as a drop-in replacement for CycleGAN on our datasets with no substantial changes. This method would also work with more than 2 subgroups per class.
>
> 7. Misspecification of the finer-grained subgroup labels is a broader challenge for robustness that we do not directly focus on in this work. We point out a few leads: (i) The group DRO algorithm upon which our method is based found that their method is robust to mis-specification of subgroups [1, Appendix B]. (ii) The setting of how to handle subgroup robustness with *no subgroup labels* is an emerging area of robustness, see for example [2] concurrent to this work.
>
> 8. Our Celeb-A experiment considers this setting of considerable imbalance in the subgroups. As shown in Table 7 (Appendix), the blonde class in Celeb-A has 2480 examples in the female subgroup but only 180 in the male subgroup, a severe imbalance. From Table 1, we see that CAMEL improves the subgroup gap on the blonde class by 3.81%: from 38.09 for ERM, 11.88 for GDRO to only 8.07 for CAMEL.
>
> 9. The numbers in parentheses are standard deviations, in the same format as Table 2. **We have also clarified this in Table 5 in the revised draft.**
>
> [1] Sagawa et al. “Distributionally robust neural networks for group shifts:  On the importance of regularization for worst-case generalization.” ICLR 2020. \
> [2] Sohoni et al. “No Subclass Left Behind: Fine-Grained Robustness in Coarse-Grained Classification Problems.” NeurIPS 2020.

---

> > ### Comment · AnonReviewer3 · 2020-11-22
> > **Response addresses comments**
> >
> > Thanks to the  authors for their  response. It addresses my comments well.

---

### Author Response · Authors · 2020-11-21
**Revision Summary -- thanks to all reviewers for thorough and insightful feedback**

We were pleased to see that reviewers found the problem **interesting and important** _(R1, R3)_, the paper **well written** _(R1, R3)_, and that model patching is **well motivated** _(R4)_, **intuitive** _(R1)_, and **sound** _(R2)_. The reviewers also agreed that the **experiments & ablations are sufficiently extensive** _(R2, R4)_ and **results compelling** _(R1, R2, R3, R4)_. We are glad that _R3_ agrees that our work would be **easy to reproduce** using the code and appendices we provide.

We have addressed the reviewers’ comments and concerns in individual responses to each reviewer. The reviews allowed us to improve our draft and the changes made in the revised draft are summarized below:

- _[R1, R3, R4]_ Updated Section 2 to clarify the setting where subgroups overlap or are shared by classes.
- _[R1, R3]_ Updated wording in introduction to clarify that subgroups are known.
- _[R2]_ Added baseline experiment on MNIST-Correlation to directly predict (Y, Z) pairs using ERM & GDRO in Section 4.2.3 and Table 5.
- _[R2]_ Added Figures 8, 9 & 10 in the Appendix with examples of CycleGAN image generations from MNIST-Correlation, CelebA and Waterbirds.
- _[R3]_ Added discussion on related work for subgroup fairness in Appendix B.5.
- _[R3]_ Improved wording before Theorem 1.
- _[R3]_ Updated Table 6 (earlier Table 5) caption to clarify the use of parentheses.
- _[R4]_ Added consistency loss ablation experiment on MNIST-Correlation using CAMEL and heuristic augmentations in Appendix D.6.2 and Table 9.

---

### Decision · Program_Chairs · 2021-01-07
**Final Decision**

**Decision:**

Accept (Poster)

**Comment:**

This paper presents an approach for mitigating subgroup performance gap in images in cases when a classifier relies on subgroup specific features. The authors propose a data augmentation approach, where synthetically produced examples (by GANs) act as instantiations of the real samples in all possible subgroups. By matching the predictions of original and augmented examples, the prediction model is forced to ignore subgroup differences encouraging invariance. The proposed method of ‘controlled data augmentations’ (as precisely called by R4) is relevant and well-motivated, the theoretical justifications support the main claims, and the experimental results are diverse and demonstrate merits of the proposed approach. As rightly pointed out by R3, ‘The appendices are also very thorough, and the code is organized well’.

In the initial evaluation, the reviewers have raised (in unison) concerns regarding overlapping subgroups per class, and an imbalance problem in the subgroups when training GANs. There were also questions reg. theoretical justifications, and empirical evaluations of the baseline methods. The authors have addressed all major concerns in the rebuttal. Pleased to report that based on the author respond with extra experiments and explanations, R2 has raised the score from 6 to 7. In conclusion, all four reviewers were convinced by the author’s rebuttal, and AC recommends acceptance of this paper – congratulations to the authors!

There is a colossal effort in the community addressing a goal similar to this work – learning invariant representations w.r.t. sensitive features by means of algorithmic fairness methods. (R1 and R3 relate to it). When preparing the final version, the authors are encouraged to elaborate more on the discussion/comparison to fairness-based methods, ideally including empirical evidence where possible (where subgroups overlap, e.g. CelebA). The AC believes this will strengthen the final revision and will have an even broader impact in the community.